# Clinical resistance to crenolanib in acute myeloid leukemia due to diverse molecular mechanisms

Haijiao Zhang[1,2], Samantha Savage[1,2], Anna Reister Schultz [1,2], Daniel Bottomly[4], Libbey White [4], Erik Segerdell[4], Beth Wilmot [4], Shannon K. McWeeney [4], Christopher A. Eide[2,3], Tamilla Nechiporuk[1,2], Amy Carlos[5], Rachel Henson[5], Chenwei Lin[5], Robert Searles[5], Hoang Ho[6], Yee Ling Lam[6], Richard Sweat[6], Courtney Follit[6], Vinay Jain[6], Evan Lind[7], Gautam Borthakur[8], Guillermo Garcia-Manero[8], Farhad Ravandi[8], Hagop M. Kantarjian[8], Jorge Cortes[8], Robert Collins [9], Daelynn R. Buelow[10], Sharyn D. Baker[10], Brian J. Druker [2,3] & Jeffrey W. Tyner[1,2]

*FLT3* mutations are prevalent in AML patients and confer poor prognosis. Crenolanib, a potent type I pan-FLT3 inhibitor, is effective against both internal tandem duplications and resistance-conferring tyrosine kinase domain mutations. While crenolanib monotherapy has demonstrated clinical benefit in heavily pretreated relapsed/refractory AML patients, responses are transient and relapse eventually occurs. Here, to investigate the mechanisms of crenolanib resistance, we perform whole exome sequencing of AML patient samples before and after crenolanib treatment. Unlike other FLT3 inhibitors, crenolanib does not induce *FLT3* secondary mutations, and mutations of the FLT3 gatekeeper residue are infrequent. Instead, mutations of *NRAS* and *IDH2* arise, mostly as *FLT3*-independent subclones, while *TET2* and *IDH1* predominantly co-occur with *FLT3*-mutant clones and are enriched in crenolanib poor-responders. The remaining patients exhibit post-crenolanib expansion of mutations associated with epigenetic regulators, transcription factors, and cohesion factors, suggesting diverse genetic/epigenetic mechanisms of crenolanib resistance. Drug combinations in experimental models restore crenolanib sensitivity.

[1] Department of Cell, Developmental & Cancer Biology, Oregon Health & Science University Knight Cancer Institute, Portland 97239 OR, USA. [2] Division of Hematology and Medical Oncology, Oregon Health & Science University Knight Cancer Institute, Portland 97239 OR, USA. [3] Howard Hughes Medical Institute, Portland 97239 OR, USA. [4] Division of Bioinformatics and Computational Biology, Department of Medical Informatics and Clinical Epidemiology, Oregon Health & Science University Knight Cancer Institute, Portland 97239 OR, USA. [5] Integrated Genomics, Knight Cancer Institute, Oregon Health & Science University Knight Cancer Institute, Portland 97239 OR, USA. [6] AROG Pharmaceuticals, Dallas 75240 TX, USA. [7] Molecular Microbiology & Immunology, Oregon Health & Science University Knight Cancer Institute, Portland 97239 OR, USA. [8] The University of Texas MD Anderson Cancer Center, Houston 77030 TX, USA. [9] University of Texas Southwestern Medical Center, Dallas 75390 TX, USA. [10] The Ohio State University College of Pharmacy and Comprehensive Cancer Center, Columbus 43210 OH, USA. Correspondence and requests for materials should be addressed to J.W.T. (email: tynerj@ohsu.edu)

Acute myeloid leukemia (AML) is a group of heterogeneous hematologic malignancies characterized by numerous cytogenetic and molecular alterations. Activating mutations in the *FMS-like tyrosine kinase 3* (*FLT3*) gene represents the most frequent molecular abnormality in AML[1,2]. The majority of the mutations in *FLT3* are internal tandem duplications (ITD), which are identified in approximately 30% of AML patients and are associated with a higher propensity for disease relapse and a shorter overall survival[3,4], even after stem cell transplantation[5]. *FLT3* point mutations in the activation loop of the tyrosine kinase domain (TKD), predominantly at residue D835, are found in an additional 7% of patients with uncharacterized prognosis[6,7].

A growing number of small-molecule FLT3 tyrosine kinase inhibitors (TKIs) have been evaluated in preclinical experiments and clinical trials, but only one agent (midostaurin) has been recently approved for this specific use. Many of the first-generation FLT3 inhibitors including midostaurin, lestaurtinib, sunitinib and sorafenib have been limited by their suboptimal efficiency and sustainability as a single drug therapy[8,9]. However, recent clinical trials with some of these agents, notably midostaurin, have revealed durable improvements in patient outcomes when administered at diagnosis in combination with standard of care chemotherapy[10,11]. The second-generation inhibitors, including quizartinib, pexidartinib, gilteritinib and crenolanib, have demonstrated enhanced potency and selectivity when administered as single-agent therapies[12–18]. Compared to other FLT3 TKIs, crenolanib demonstrates several appealing characteristics to target *FLT3* mutations in AML. As a potent type I pan-FLT3 inhibitor, crenolanib retains activity against *FLT3* TKD mutations[19], which have been shown to be the major resistance mechanisms for quizartinib and sorafenib[20–24]. Therefore, crenolanib is a candidate therapy for de novo AML patients with *FLT3* TKD mutations as well as relapsed patients with TKD mutations acquired after treatment with other FLT3 TKIs[25].

Crenolanib has been evaluated in two phase II clinical trials in chemotherapy or TKI refractory/relapsed AML patients with *FLT3* mutations. Cumulatively, a high response rate (complete response with incomplete blood count recovery (CRi) of 37%, and partial response (PR) of 11% in prior TKI-naive group; 15% complete response (CR)/CRi and 13% PR in prior TKI group) was achieved with crenolanib single-agent therapy.[26] Details of the clinical trials are reported elsewhere[14,25,26]. However, similar to other FLT3 TKIs observed in early clinical trials, despite initial response, subsequent drug resistance and disease relapse occurred in the majority of patients[8,9,14,25,26]. We, therefore, performed whole exome sequencing (WES) and targeted deep sequencing on a series of samples from crenolanib-treated patients to investigate the relationship between drug resistance and genetic signatures (data can be explored and visualized in our Vizome, online data browser (www.vizome.org)). We were initially interested in investigating whether crenolanib resistance followed similar mechanisms as other FLT3 TKIs (quizartinib, gilteritinib and sorafenib)[27–30], where secondary *FLT3* mutations in the activation loop and/or gatekeeper residue play a major role. Given the nature of heterogeneous genetic alterations and selective pressure of chemotherapy and prior TKI treatment in relapsed/refractory AML patients on these trials, we also aimed to characterize the impact of co-occurring clones or subclones with other somatic mutations on crenolanib response and disease recurrence.

We observed that crenolanib-resistant *FLT3* secondary mutations (one patient with K429E mutation and two patients with gatekeeper mutations) are infrequent. The majority of patients exhibited a diverse spectrum of mutations associated with chromatin modifiers, cohesion, spliceosomes and transcription factors, which mostly expanded during treatment, suggesting an elaborate genetic/epigenetic mechanism of resistance to crenolanib.

## Results

**FLT3 secondary mutations are infrequent.** We first determined whether secondary *FLT3* mutations were acquired during treatment by sequencing available patient samples obtained after at least 28 days of crenolanib treatment as well as baseline samples obtained before crenolanib treatment initiation. Consistent with previous reports, no de novo activation loop mutations were detected in *FLT3*-ITD patients after crenolanib treatment as determined from 18 *FLT3*-ITD patients sequenced by exome sequencing (mean ± standard error of the mean (SEM) coverage is 177 ± 24) and 6 *FLT3*-ITD patients sequenced by Miseq (coverage 232,809 ± 9388)[31]. Variant allele frequencies (VAFs) of *FLT3* D835 or *FLT3*-ITD were eliminated or cleared in 11 out of 21 or 11 out of 39 patients, respectively; and maintained or expanded in the rest of patients (Supplementary Data 1-2). *FLT3* F691 mutations were detected in two patients (Fig. 1a). Both patients were previously treated with quizartinib and one patient harbored 17% VAF of *FLT3* F691L prior to crenolanib treatment. Consistent with previous studies[14,20,23,25,26], low VAFs of non-D835 *FLT3* secondary mutations A833S, D839Y/G, N841K and Y842C and a small insertion/deletion (R834D835I836 (RDI->RP)) were detected at baseline and eliminated during the course of crenolanib treatment (Fig. 1a and Supplementary Table 1). Four *FLT3* point mutations (D200N, K429E, Y572C and L601F) were maintained after crenolanib treatment in four individual patients. Y572C was previously reported to be transforming[32].

To characterize the leukemogenic and drug-resistant potential, *FLT3* wild-type (WT) and point mutations were introduced into Ba/F3 cells and evaluated by interleukin-3 (IL-3) withdrawal assay and drug sensitivity profiling. As expected, FLT3 D835Y (positive control) and Y572C transformed Ba/F3 cells (Fig. 1b). FLT3 K429E was also observed to transform Ba/F3 cells, although with slower kinetics compared to FLT3 D835Y and Y572C (Fig. 1b). The other two mutations (D200N and L601F) and the gatekeeper mutation F691L did not transform Ba/F3 cells.

In drug profiling assays, FLT3 K429E expressing Ba/F3 cells demonstrated reduced crenolanib sensitivity compared to Molm14 cells and Ba/F3-FLT3 D835Y, whereas FLT3 Y572C expression in Ba/F3 cells had no effect on the crenolanib dose response curve compared with FLT3 D835Y cells (Fig. 1c). Accordingly, bone marrow (BM) cells expressing FLT3 K429E were less sensitive to crenolanib compared with cells expressing FLT3 D835Y (Supplementary Figure 1a). To determine whether *FLT3* mutations that did not transform Ba/F3 cells could confer resistance to crenolanib, we generated double mutant *FLT3*-D200N/D835Y, K429E/D835Y, L601F/D835Y and F691L/D835Y (Fig. 1d). As expected, these mutations all transformed Ba/F3 cells (Supplementary Figure 1b). Consistent with the FLT3 K429E transformed cells, FLT3 K429E/D835Y transformed cells showed decreased crenolanib sensitivity, although it did not confer the same degree of crenolanib resistance as the gatekeeper, *FLT3* F691L/D835Y mutation (Fig. 1d, e). FLT3 D200N/D835Y and FLT3 L601F/D835Y both showed similar crenolanib sensitivity as FLT3 D835Y (Fig. 1d, e) suggesting that these mutations are passenger mutations and do not contribute to crenolanib resistance. This is also consistent with their presence prior to crenolanib therapy and VAFs that were unchanged over the course of therapy (Fig. 1a).

**Differential mutation profiles in prior TKI-treated patients.** Since prior FLT3 inhibitor treatment was also identified as an adverse prognostic factor of crenolanib clinical response[26], we compared the mutational profiles of these patients (pre-TKI) to patients who had not received prior TKI therapy (TKI naive) (Supplementary Table 2-3 and Supplementary Data 3-4).

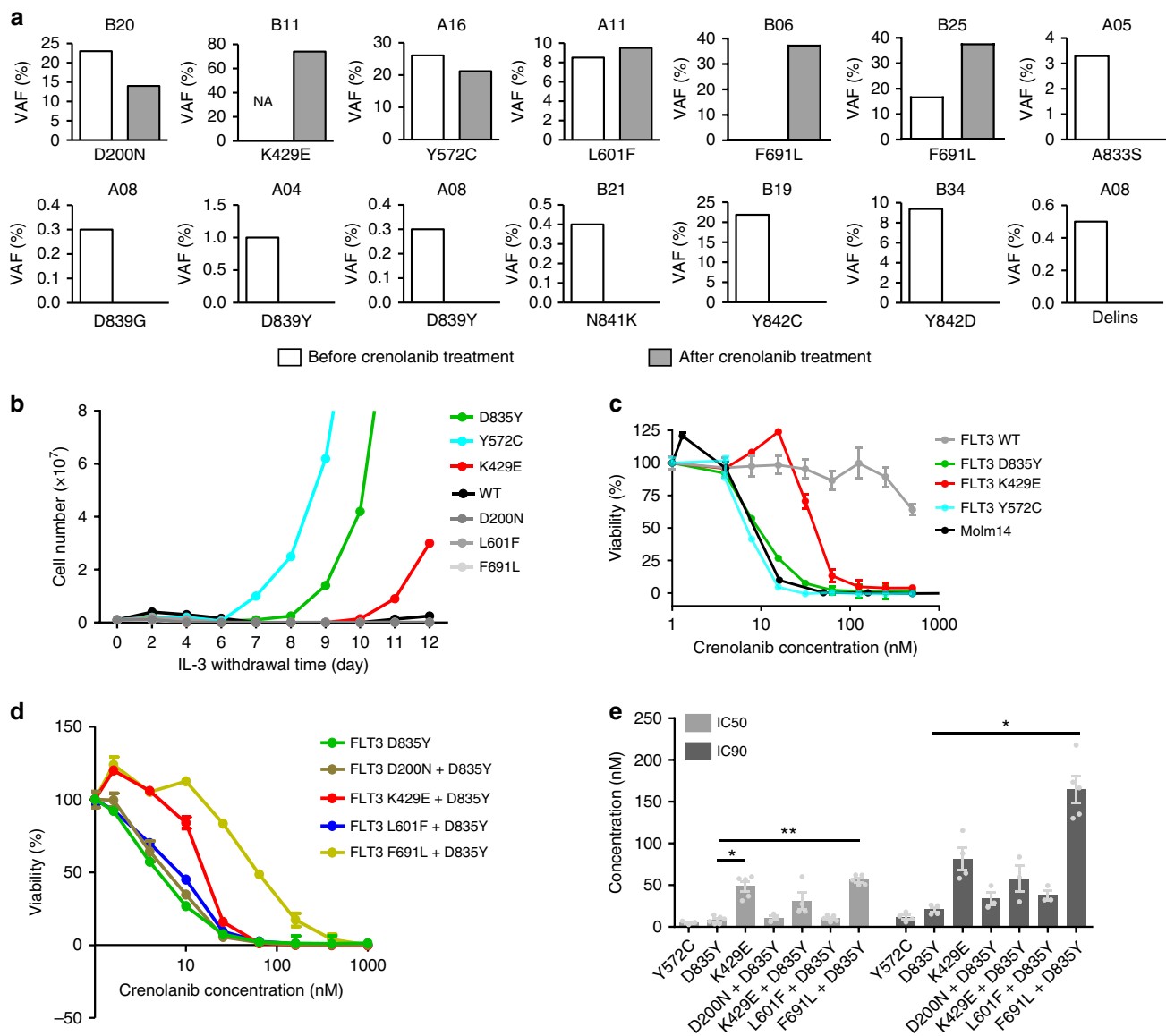

**Fig. 1** *FMS-like tyrosine kinase 3* (*FLT3*) K429E demonstrates reduced crenolanib sensitivity. **a** Variant allele frequency (VAF) of non-D835 *FLT3* mutations during crenolanib treatment. Low VAFs of FLT3 A833S, D839Y/G, N841K, Y842C/D and delIns were detected at baseline, and these mutation clones were eliminated during the course of crenolanib treatment. *FLT3* F691L mutations were detected in two patients previously treated with quizartinib. Four *FLT3* mutations (D200N, K429E, Y572C and L601F) were identified at the time of treatment termination in four individual patients. DelIns: R834D835I836 (RDI-−>RP). **b** FLT3 K429E transforms Ba/F3 cells. Ba/F3 cells expressing empty vector, FLT3 wild-type (WT) and mutants were grown in medium without interleukin-3 (IL-3) and cells were counted every other day for 12 days. **c**, **d** Ba/F3 cells expressing FLT3 K429E and FLT3 K429E/D835Y demonstrate reduced crenolanib sensitivity. Graphs depict mean ± SEM of cell viabilities of Ba/F3 cells expressing empty vector, FLT3 WT or mutants treated with dose gradients of crenolanib for 72 h determined by MTS. **e** Mean ± SEM of crenolanib half-maximal inhibitory concentration (IC$_{50}$) and 90% inhibitory concentration (IC$_{90}$) values of Ba/F3 cells transformed with FLT3 WT and mutants as presented in **c**, **d**. Graphs and images shown are representatives from six experiments. Statistical significance was assessed using one-way analysis of variance (ANOVA) and Kruskal–Wallis test comparing each condition to the respective *FLT3* D835Y and expressed as: *$p < 0.05$; **$p < 0.01$

Available data from the targeted gene panel and/or WES of 50 patients before crenolanib treatment were analyzed and recurrent variants within 24 commonly mutated genes were detected (Fig. 2a, b and Supplementary Data 1). The average number of pathogenic variants was highest among the pre-TKI group, while similar frequencies were observed between the TKI-naive and the de novo AML cases reported in The Cancer Genome Atlas (TCGA; Fig. 2d and Supplementary Table 8). Consistent with previous studies[24], higher frequencies of *FLT3*-ITD and *FLT3* TKD combination mutations were observed in the pre-TKI group (Supplementary Figure 2). Furthermore, the frequency of *NRAS,*

*KRAS, RUNX1, IDH1, WT1, TET2* truncation and *ASXL1* mutations were higher in the pre-TKI group (Fig. 2b).

**Differential mutation profiles in crenolanib poor responders**. We next compared the coexisting mutation profiles between different crenolanib response groups. We observed that the average number of mutations at trial entry was higher for crenolanib poor responders (hematological improvement (HI)+resistant disease (RD)) compared to crenolanib good responders (CR/CRi +PR) (Fig. 2c). Furthermore, higher frequencies of *NRAS, TET2, IDH1, IDH2, U2AF1, STAG2, KRAS, CSF3R, TET2* truncation

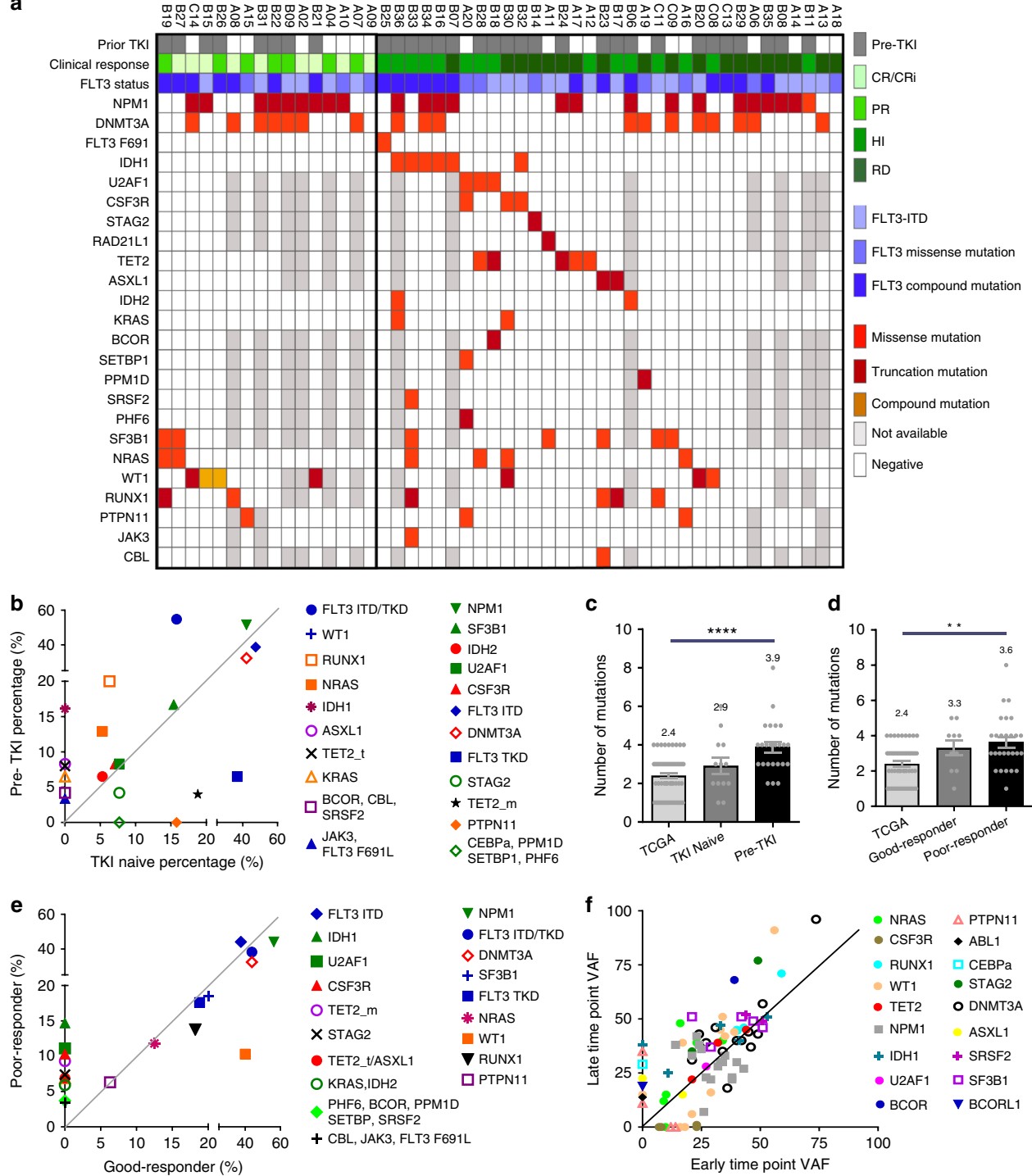

**Fig. 2** Differential mutation profiles in pre-tyrosine kinase inhibitor (TKI)-treated patients and crenolanib poor responders. **a** Mutation spectrum identified by whole exome sequencing and targeted deep sequencing before crenolanib treatment in TKI-naive and pre-TKI group. **b** Comparison of frequencies of specific mutated genes detected in TKI-naive, prior TKI treatment patients. **c** The graph depicts mean ± SEM of mutation numbers discovered in The Cancer Genome Atlas (TCGA) de novo acute myeloid leukemia (AML) patients with *FMS-like tyrosine kinase 3* (*FLT3*) mutations ($n = 56$, mutation number range 1–4), TKI-naive ($n = 13$, mutation number range 1–6) and prior TKI ($n = 24$, mutation number range 2–8) treatment patients. **d** The graph depicts mean ± SEM of number of mutations in TCGA de novo AML patients, crenolanib good responders ($n = 10$, mutation number range 1–5) and poor-responders ($n = 27$, mutation number range 1–8). For (**c**, **d**), *FLT3*-ITD and *FLT3* tyrosine kinase domain (TKD) compound mutation is counted as 2. **e** Comparison of frequencies of specific mutated genes detected in crenolanib good responders and poor responders. **f** Graph depicts variant allele frequency (VAF) change of gene mutations during crenolanib treatment. Statistical significance was assessed using one-way analysis of variance (ANOVA) and Kruskal–Wallis test comparing each condition to the respective *FLT3* D835Y and expressed as: **p < 0.01, ****p < 0.0001

and *ASXL1* mutations were present in poor responders compared with crenolanib good responders prior to crenolanib treatment (Fig. 2e, Supplementary Table 4 and Supplementary Figure 3). We did not observe an association of *FLT3* mutation status (ITD vs. TKD vs. ITD+TKD) with crenolanib response (Supplementary Table 4 and Supplementary Figure 3). Changes in VAF were then analyzed from available paired samples to identify new or expanded clones during crenolanib treatment. We observed that the majority of *TET2*, *DNMT3A*, *RUNX1*, *U2AF1*, *SF3B1* and *IDH1* mutations were present with approximately 50% VAF, and the VAFs remained unchanged during crenolanib treatment (Fig. 2f), indicating they were present in the founder clone and may not be sensitive to crenolanib treatment. VAFs of variants of *NRAS*, *BCOR*, *STAG2*, *CEBPA* and *ASXL1* increased during crenolanib treatment, suggesting these mutations may contribute to a clonal selection mechanism of drug resistance. We observed both the expansion and reduction of *NPM1* and *WT1* mutations during crenolanib treatment, indicating these mutations alone may not be sufficient to render drug resistance. Interestingly, loss of heterogeneity and VAF expansion were observed in two *DNMT3A*, one *WT1*, and one *RUNX1* mutation at crenolanib resistance, suggesting they might contribute to disease resistance. Surprisingly, VAF of *CSF3R* (5 out of 5) and *JAK3* (1 out of 1) and *PTPN11* (2 out of 4) mutations dropped to undetectable levels during crenolanib treatment (Fig. 2f). Of note, multiple other gene mutations were present in these samples. There might be two possible explanations for this phenomenon: crenolanib may be able to inhibit *CSF3R*, *JAK3* and *PTPN11* mutation clones due to targeting kinases other than FLT3 (e.g., Janus kinases (JAKs), mitogen-activated protein kinase (MAPK) pathway); or *CSF3R*, *JAK3* or *PTPN11* mutation clones were suppressed by other more aggressive outgrowth clones with differing mutational profiles. Since crenolanib targets platelet-derived growth factor receptor (PDGFR) and FLT3 with high specificity and there are two patients clearly showing *PTPN11* mutations expansion at disease relapse (A14 and B27), the second model appears more likely. Nonetheless, these data indicate that *CSF3R* and *JAK3* mutations do not harbor high growth advantage during crenolanib treatment in the presence of other clones, and mutations in these genes are unlikely to confer crenolanib resistance, whereas *PTPN11* mutations could be repressed or acquired during crenolanib treatment and contribute to drug resistance.

**Subclonal RAS mutations contribute to crenolanib resistance.** Since mutations in RAS signaling pathway genes (*NRAS*, *PTPN11*, *KRAS* and *CBL*) are enriched in crenolanib poor responders and/or pre-TKI group (Fig. 2b, e), and/or expanded or acquired during crenolanib treatment (Fig. 2f), it is likely they can confer crenolanib resistance.

To determine whether these mutations co-occur with *FLT3* mutations within the same clone or occur in an independent *FLT3* WT clone, we compared the absolute VAF and the direction of VAF change of these gene variants with respect to absolute VAF and VAF changes of the *FLT3* mutations. For example, if the VAF of the RAS pathway gene and the *FLT3* mutations are both greater than 50%, or if a lower baseline VAF of the RAS pathway mutation changes in the same trajectory as the *FLT3* mutation after crenolanib therapy, we can assume that they have a high probability of occurring within the same leukemic clone. In this manner, we observed that the majority of the *NRAS* and *KRAS* pathway mutations were present in independent clones not harboring the *FLT3* mutations (Fig. 3a). However, we did observe that one *NF1* and two *CBL* mutations co-occurred with *FLT3* mutations in the same clone. Three out of four *PTPN11* mutations (A14, A16 and A15) appear to co-occur with *FLT3* TKD or ITD

mutations. We, therefore, analyzed the drug response profile of cells harboring *FLT3* and RAS pathway compound mutations by introducing FLT3 D835Y and PTPN11 A72D into Ba/F3 cells.

Significantly increased crenolanib half-maximal inhibitory concentration ($IC_{50}$) and 90% inhibitory concentration ($IC_{90}$) were observed in PTPN11 A72D/FLT3 D835 co-transduced cells relative to PTPN11 WT/FLT3 D835 cells (Fig. 3b, c). Consistently reduced crenolanib sensitivity was observed in MV4–11 cells expressing PTPN11 A72D compared to PTPN11 WT and controls (Supplementary Figure 2a). To circumvent the compromised drug sensitivity of cells harboring these compound mutations, we combined crenolanib with the MEK inhibitor, trametinib. PTPN11 WT/FLT3 D835 and PTPN11 A72D/FLT3 D835 cells did not respond to trametinib single-agent exposure at low concentrations (Supplementary Figure 2b-c). However, a synergistic effect of trametinib in combination with crenolanib was observed against PTPN11 A72D/D835Y cells and to a lesser degree against FLT3 D835Y only or PTPN11 WT/FLT3 D835 cells (Fig. 3b, c and Supplementary Figure 4b-d).

**Concomitant *TET2* mutations confer crenolanib resistance.** Five patients with *TET2* frameshift/nonsense mutations demonstrated adverse prognosis, whereas four other patients with *TET2* missense mutations did not show a higher incidence of unfavorable response to crenolanib (Fig. 4a and Supplementary Table 6), suggesting that *TET2* truncation mutations may contribute to crenolanib resistance. VAF analysis demonstrated co-occurrence and persistence of *TET2* and *FLT3* mutations, indicating up-front drug resistance for cases harboring *FLT3/TET2* compound mutations at the start of therapy. These findings are consistent with recent work showing *TET2* mutations contribute to quizartinib resistance in mouse models[33]. To further test this hypothesis, we performed a drug sensitivity assay with BM stem/progenitor cells from *Flt3*-ITD knock-in (*Flt3*[ITD]) and *Flt3*-ITD knock-in/*Tet2* knockout mice (*Flt3*[ITD]; *Tet2*[+/−]). We observed that *Flt3*[ITD]; *Tet2*[+/−] progenitor cells did not respond to crenolanib at 10 or 100 nM, whereas *Flt3*[ITD] cells responded well at these drug concentrations (Fig. 4b–d). These data confirmed that FLT3-ITD, in concert with TET2 loss of function, is resistant to the single-agent FLT3 inhibitor. Excitingly, *Flt3*[ITD]; *Tet2*[+/−] cells demonstrated sensitivity to azacytidine, and cell differentiation analysis demonstrated a clearance of Sca1+ progenitor cells with azacytidine in both *Flt3*[ITD] and *Flt3*[ITD]; *Tet2*[+/−] mouse BM stem/progenitor cells (Supplementary Figure 3).

**IDH1 and IDH2 mutations contribute to crenolanib resistance.** Higher frequencies of *IDH1* and *IDH2* mutations were observed in the pre-TKI and crenolanib poor responder groups (Fig. 2b–f), indicating *IDH1* and *IDH2* mutations might be associated with TKI treatment resistance. *IDH1* mutations co-occurred in the same clone with *FLT3*-ITD/TKD mutations in five patients (B01, B07, B34, B32 and B16), whereas the *IDH2* mutations were highly likely to be in *FLT3*-ITD/TKD independent clones in two patients (A08 and B04) and possibly co-occurred with *FLT3*-ITD in one patient (B06) (Fig. 5a). To investigate whether co-occurring *IDH1* mutations perturb crenolanib sensitivity, we performed drug profiling of cells harboring both mutations. We did not observe decreased crenolanib sensitivity in FLT3-ITD stem cells transduced with IDH1 R132H compared to FLT3-ITD cells transduced with IDH1 WT (Fig. 5b). We observed that cells harboring IDH1 R132H/FLT3 D835Y demonstrated similar crenolanib sensitivity as IDH1 WT/D835Y cells ($16.2 \pm 8.0$ nM vs. $18.9 \pm 13.9$ nM) (Fig. 5c). Consistent with previous studies, an IDH1 inhibitor as a single agent did not impact on cell viability. However, the IDH1 inhibitor (AG5198) did enhance crenolanib sensitivity (Fig. 5d and

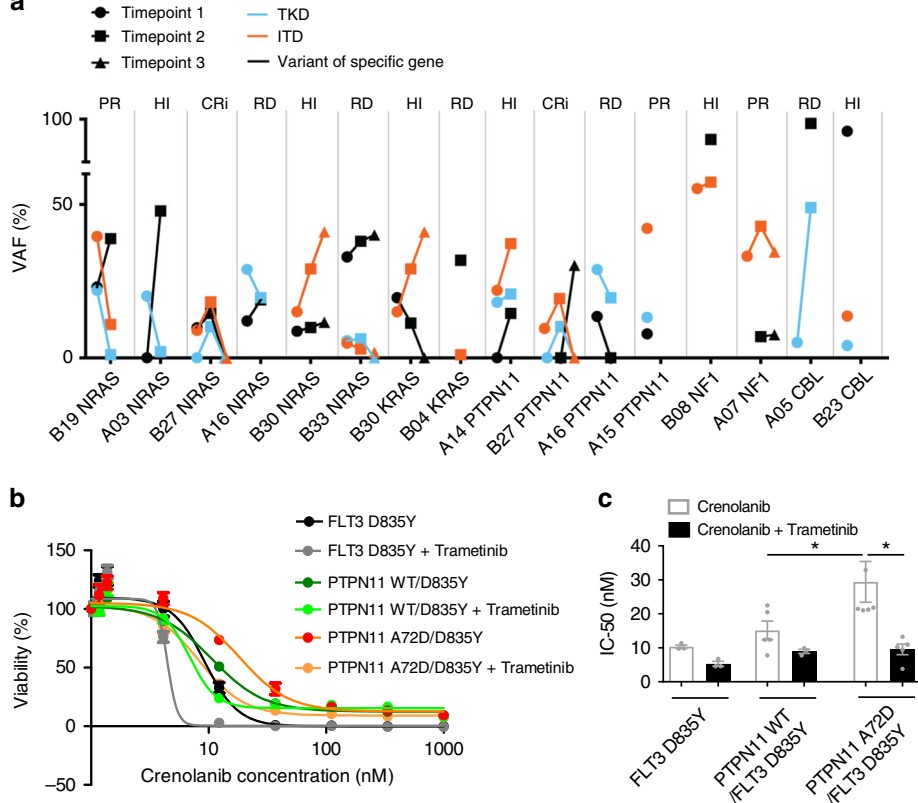

**Fig. 3** RAS pathway mutations contribute to crenolanib resistance and disease relapse. **a** The graph depicts variant allele frequencies (VAFs) of *FLT3*-ITD/ TKD and RAS pathway mutations during crenolanib treatment. **b** Graph depicts higher mean ± SEM of cell viabilities of crenolanib-treated PTPN11 A72D/ FLT3 D835Y Ba/F3 cells in comparison to PTPN11 WT/ FLT3 D835Y co-expressing Ba/F3 cells and *FLT3* D835Y-alone expressing Ba/F3 cells determined by MTS assay. **c** Graph depicts mean ± SEM of crenolanib half-maximal inhibitory concentration (IC$_{50}$) in (**b**). Data shown are from five biological replicates. Statistical significance was assessed using one-way analysis of variance (ANOVA) together with Dunn's multiple comparisons tests and expressed as: *$p < 0.05$

Supplementary Figure 6). Further validation from similar clinical trials and from in vivo experiments studying TKI sensitivity in *Idh1* mutation and *Flt3*-ITD double knock-in mice are needed.

**Concomitant *TP53* mutations confer crenolanib resistance.** *TP53* pathway mutations are mutually exclusive with *FLT3* mutations in de novo AML and were shown to be related to FLT3 inhibitor resistance[34,35]. Interestingly, we observed one *TP53* mutation and one *PPM1D* mutation that co-occurred with *FLT3*-ITD and/or *FLT3* TKD in two individual patients with crenolanib poor response. To test whether TP53 loss of function confers crenolanib resistance, we performed a knockout of TP53 expression in Molm13 cells using the clustered regularly inter-spaced short palindromic repeats (CRISPR)/CRISPR-associated 9 (Cas9) system (Supplementary Figure 7). We observed significant reduced crenolanib sensitivity (Fig. 5e–g) of TP53 knockout cells. Further drug combinations involving FLT3 TKI- and TP53-based cancer therapy are warranted in patients harboring *FLT3* and *TP53* combination mutations.

**Mutation spectrum and clonal pattern of crenolanib-treated samples.** Overall in this crenolanib-treated patient cohort, 5.9% (3/51) of patients demonstrated *FLT3* secondary mutations; 9.8% (5/51) of patients showed concomitant *TET2* truncation muta- tions; 29.4% (15/51) of patients demonstrated mutations in alternative signaling pathways; 19.6% (10/51) demonstrated *IDH1/2* mutations; 3.9% (2/51) demonstrated *TP53* and *PPM1D* mutation, and another 23.5% (12/51) of patients demonstrated

cohesion, splicing factor, epigenetic and/or transcription factor mutations (Fig. 6a).

In general, the clonal evolution patterns observed from this series of patient samples could be classified into three distinct groups. The first pattern is of a primary drug refractory clone (Fig. 6b). In this pattern, *FLT3* mutations co-occur with other drug-resistant gene mutations in the same clone and the VAF of both mutations persist or increase during drug treatment. The co- occurring mutations are normally present in the founder clone, such as *TET2*, *IDH1* and *TP53* pathway mutations. In line with a previous study[33] and the data shown here, the patient presented in Fig. 6b exhibited both *FLT3*-ITD and *TET2* mutations that co- occurred, and this concomitant mutation pattern was completely resistant to crenolanib. These data support the notion that up-front drug combinations may be useful to target certain mutational patterns, such as *FLT3* and *TET2* co-occurring mutation. The second pattern is acquisition or expansion of additional mutations in the context of a *FLT3* mutation with either an original, dominant *FLT3*-mutant clone that exhibits new mutations at the time of drug resistance or a minor *FLT3*-mutant clone with additional mutations that expands during crenolanib treatment (Fig. 6c). Deep sequencing of samples at the start of FLT3 inhibitor therapy could identify problematic coexisting mutations and apply drug combinations that pre-emptively target the expanding clone. The third pattern is the acquisition of leukemic clones independent of the clone with the *FLT3* mutation (Fig. 6d). In this pattern, the VAF of *FLT3* mutations decreased, whereas drug-resistant sub- clones within a *FLT3* WT clone emerged or expanded and became

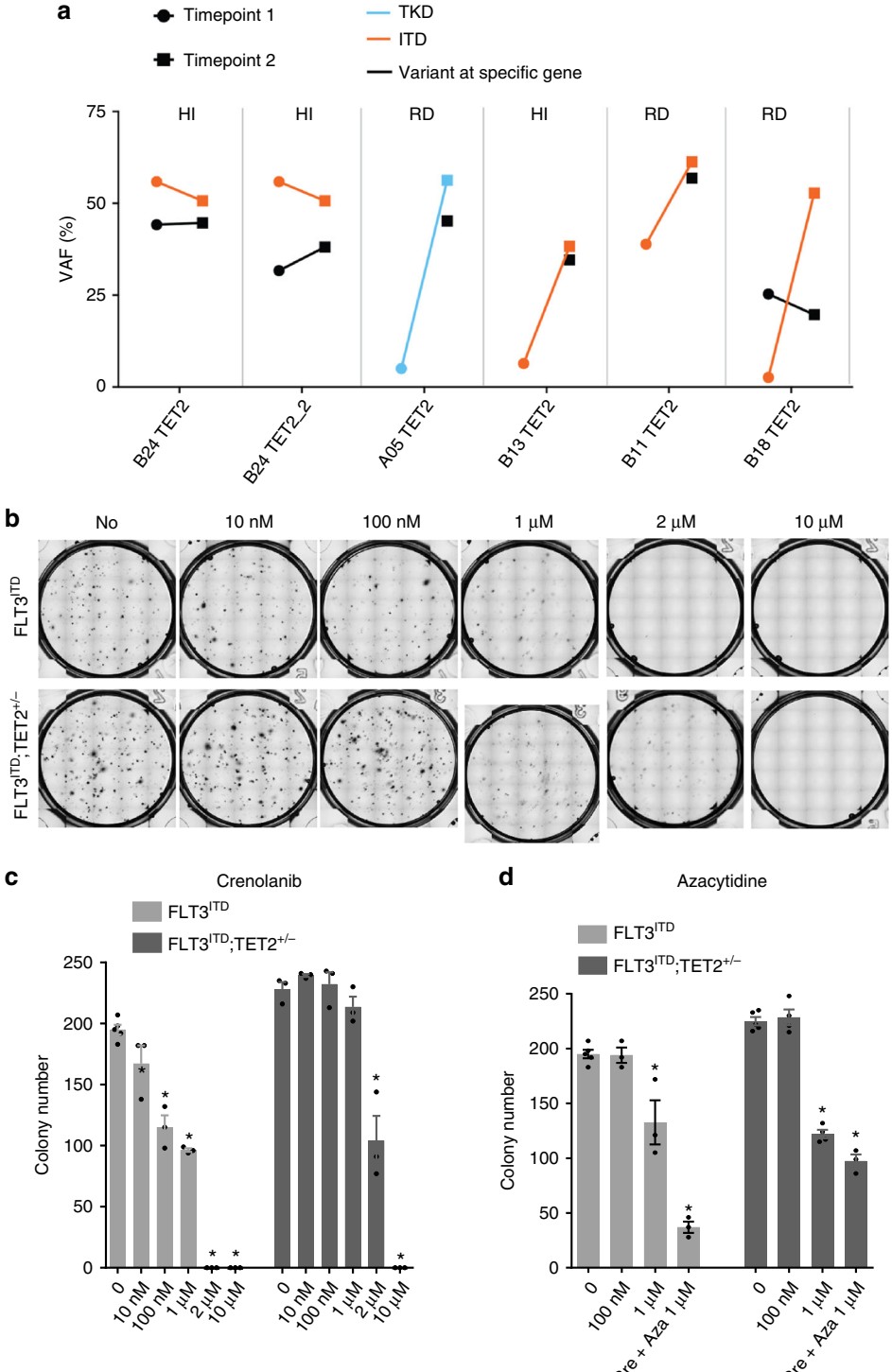

**Fig. 4** *TET2* nonsense/frameshift mutations contribute to crenolanib unresponsiveness. **a** Variant allele frequencies (VAFs) of *FLT3*-ITD/TKD and *TET2* nonsense/frameshift mutations during crenolanib treatment. **b** Representative images of colony-forming unit (CFU) assay demonstrate reduced crenolanib sensitivity of Flt3^ITD;Tet2^+/− mouse stem cells. Graph depicts mean ± SEM of colony numbers for three or four replicates of Flt3^ITD and Flt3^ITD;Tet2^+/− mouse stem cells treated with gradient concentrations of crenolanib **c** or azacytidine and two drugs in combination **d** as described in the Methods. Statistical significance was determined using two-tailed nonparametric Student's *t*-tests (Mann–Whitney test) comparing each group to the non-treated group and expressed as *$p < 0.05$

the dominant leukemic clones. This group could be observed as initial crenolanib good responders, where the decrease of the *FLT3* mutation VAF at relapse suggests that crenolanib is effective at eradicating or controlling the *FLT3*-mutant leukemic clone. Again, up-front or follow-up deep sequencing to identify potential sub-clones with resistant mutations could be helpful. Combination or sequential treatment with drugs targeting the expanding clones could prove effective in the future. In our patient cohort, *NRAS*, *STAG2*, *CEBPA*, *IDH2* and *ASXL1* mutation clones are common *FLT3*-independent drug-resistant clones.

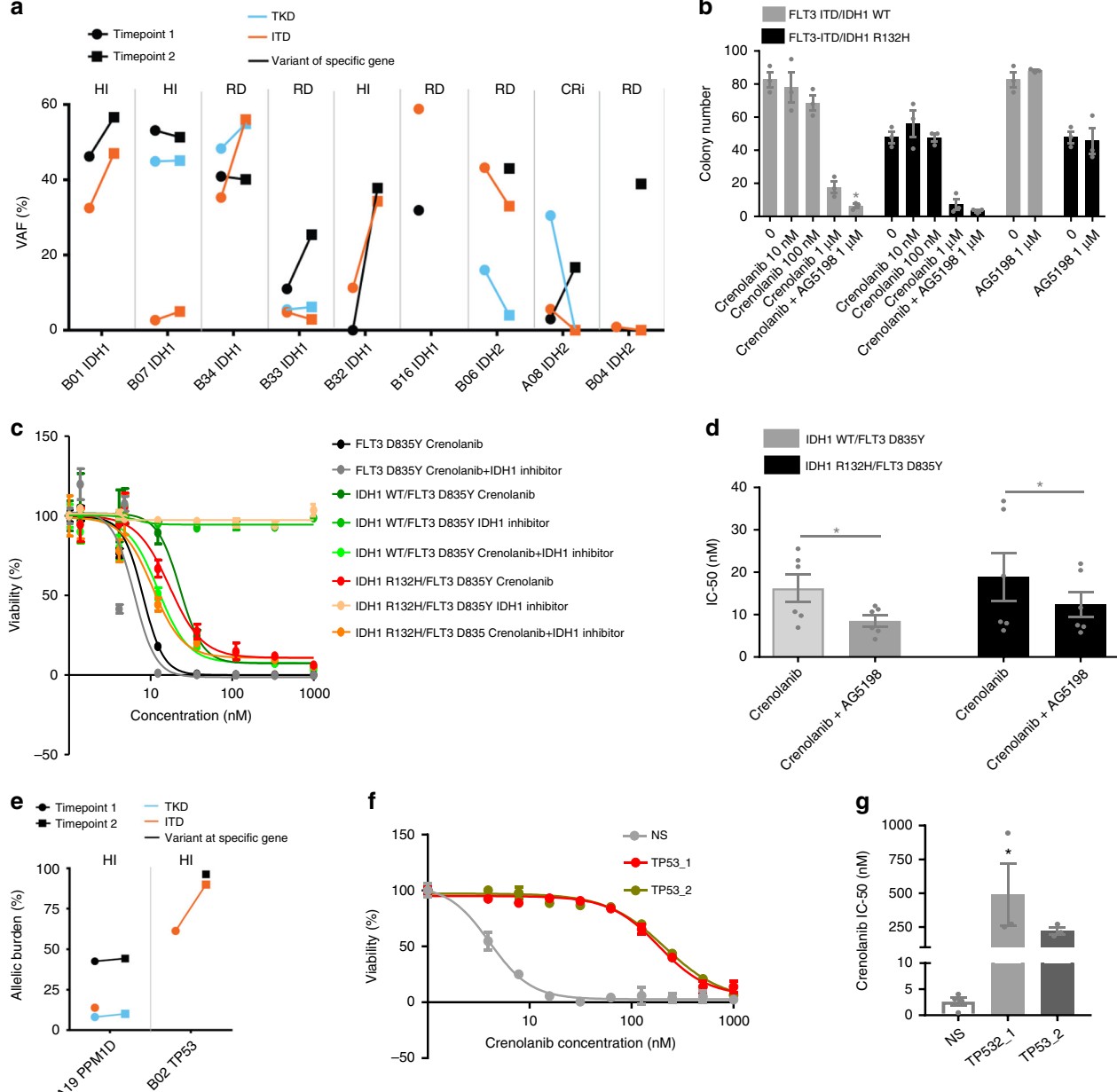

**Fig. 5** *IDH1* and *IDH2* mutations contribute to crenolanib resistance. **a** Variant allele frequencies (VAFs) of *FLT3*-ITD/TKD and *IDH1* or *IDH2* during crenolanib treatment. Notably, we did not detect the *IDH1* mutation in B32 before crenolanib treatment by exome sequencing. However, a targeted gene panel detected this *IDH1* mutation before crenolanib treatment. **b** Graph depicts mean ± SEM of colony numbers of FLT3-ITD/IDH1 WT and FLT3-ITD/IDH1 R132H expressing mouse stem cells treated with gradient concentrations of crenolanib, AG5198 or two drugs in combination. **c** Graph depicts cell viabilities of FLT3 D835Y, IDH1 WT/ FLT3 D835Y and IDH1 R132H/ FLT3 D835Y expressing Ba/F3 cells treated with crenolanib, or crenolanib in combination with IDH1 Inhibitor (AG5198) determined by MTS assay. **d** Graph depicts decreased half-maximal inhibitory concentration (IC$_{50}$) of crenolanib and AG5198 in combination compared to crenolanib alone. Statistical significance was determined using two-tailed nonparametric Student's *t*-tests (Mann–Whitney test) comparing each group to the non-treated group. *TP53* mutations co-occur with *FLT3* mutations and confer crenolanib resistance. **e** Graph depicts VAFs of *FLT3*-ITD/TKD and *TP53* or *PPM1D* mutation during crenolanib treatment. **f** Representative graph depicts higher mean ± SEM of cell viability of crenolanib-treated Molm13 cells expressing CRISPR/Cas9 and single-guide RNAs (sgRNAs) targeting *TP53* compared to cells expressing CRISPR/Cas9 and a non-specific targeting sgRNA (NS) control. TP53_1: sgRNA1 targeting *TP53*; TP53_2: sgRNA2 targeting *TP53*. **g** Graph depicts mean ± SEM of crenolanib IC$_{50}$ in (**b**). Data shown are from three or five biological replicates. Statistical significance was assessed using one-way analysis of variance (ANOVA) together with Dunn's multiple comparisons tests and expressed as: *$p < 0.05$

## Discussion

Small-molecule targeted TKIs have revolutionized cancer treatment with a fast, selective and robust response and are being increasingly used in clinical settings. Strikingly, in the current clinical trial, crenolanib achieved good responses in around 28% of patients who failed multiple other FLT3 inhibitors[14,25,26]. However, like many other TKIs, the acquisition of drug-resistant clones through selective pressure eventually led to disease relapse. A detailed understanding of the molecular changes associated with drug resistance is critical for identifying prognostic markers and additional targets to circumvent drug resistance and ultimately benefit *FLT3*-mutant AML patients with durable therapeutic responses.

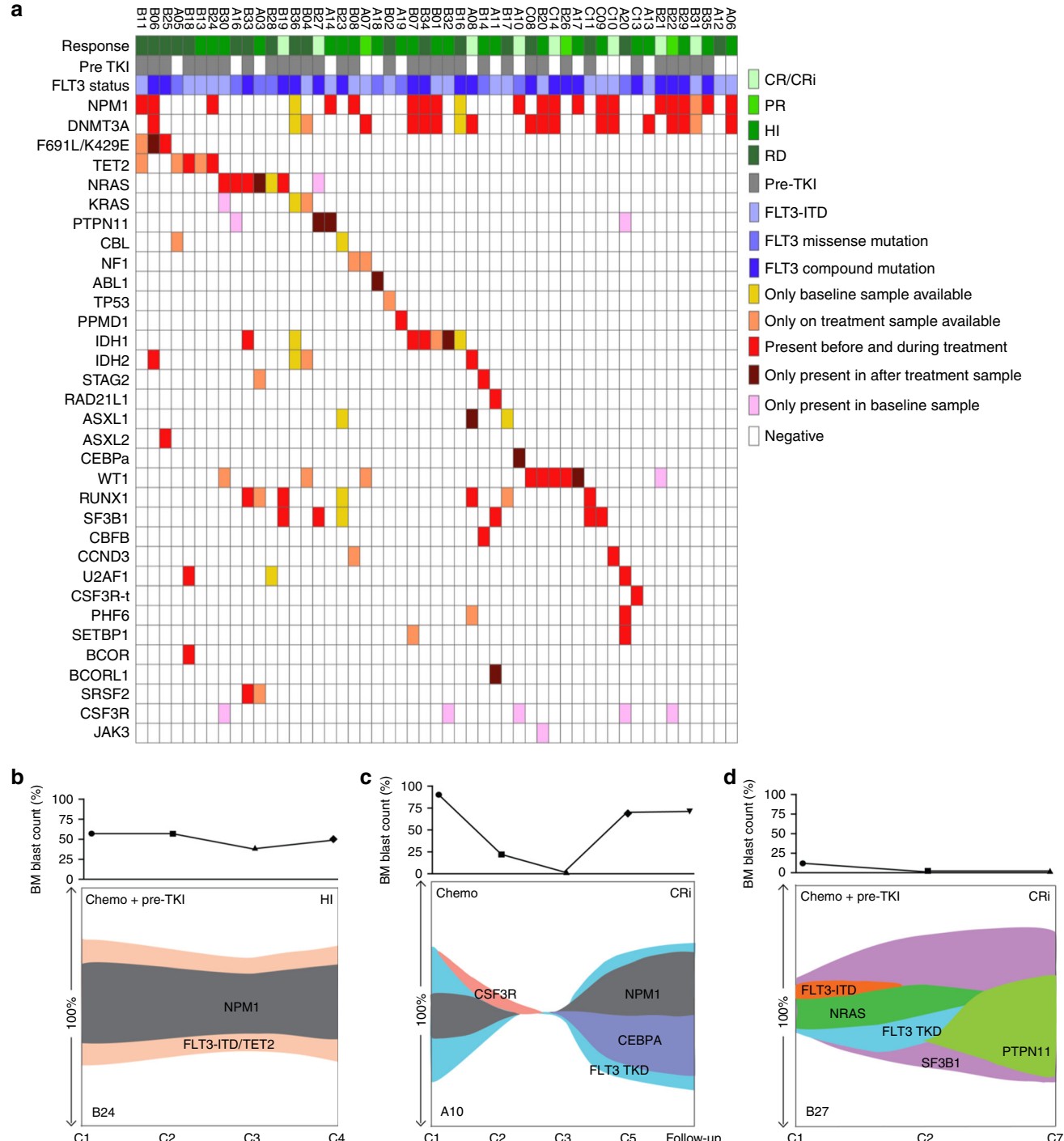

**Fig. 6** Mutation spectrum and clonal patterns of patients treated with crenolanib. **a** Graph depicts mutations identified by exome sequencing, gene panel and/or targeted sequencing. Each column displays a patient; each row denotes a specific gene. Recurrently mutated genes are color-coded for only present before crenolanib treatment, only present in after crenolanib treatment samples, persistence before and after crenolanib treatment, with no samples available before or after crenolanib treatment. **b** Graph depicts primary drug-resistant clone. In this case, *FLT3* mutation clones co-occur with a *TET2* mutation clone and the variant allele frequencies (VAFs) of the combined mutation persist during drug treatment. **c** The second pattern is the acquisition or expansion of additional mutations in the context of a *FLT3* mutation. In this case, the original, dominant *FLT3* tyrosine kinase domain (TKD) clone was inhibited by crenolanib after three cycle treatments. However, *CEBPA* mutation was acquired and expanded at the time of drug resistance in *FLT3* TKD clone. **d** The third pattern is the acquisition of subclones independent of *FLT3* mutation clones. In this case, FLT3 mutation clones were eliminated by crenolanib; however, a *PTPN11* mutation clone emerged during crenolanib treatment. Of note, the clonal patterns shown do not include mutation information before crenolanib treatment. C: crenolanib treatment cycle number; Chemo: chemotherapy. The y-axis stands for VAF

Previous studies suggest that a common drug-resistant mechanism of TKI treatment is secondary mutations of the targeted kinase itself which could be classified into two major groups[21]. The first group is TKD mutations, which are specifically resistant to type II FLT3 TKIs including quizartinib, sorafenib, ponatinib and pexidartinib[36–41]. *FLT3* N676K was first characterized to confer resistance to midostaurin and more recently also shown to contribute to resistance to quizartinib[22,42]. Point mutations at residues D835, I836 and Y842 have been shown to be predominantly associated with quizartinib and sorafenib resistance[20–22,43]. The second group is mutations on the gatekeeper residue of the targeted kinases. Gatekeeper mutations enhance the binding affinity of adenosine triphosphate (ATP) which competes with TKIs to bind to the ATP binding pocket, thereby preventing TKI binding to the kinase and rendering resistance to the TKI treatment[44]. *FLT3* F691L was shown to be resistant to the majority of FLT3 TKIs including crenolanib, but not ponatinib and pexidartinib[36,39,45]. One study showed that 3 out of 8 quizartinib-treated patients[27] and 3 out of 16 and 4 out of 20 gilteritinib-treated patients acquired a *FLT3* gatekeeper mutation[29,30]. In vitro saturation assays have been widely used to screen and identify secondary drug resistance mutations. Previous studies showed that only two recurring mutant clones out of $300 \times 10^6$ initial clones screened were retained in the presence of crenolanib (100 nM)[25]. Consistent with the in vitro saturation assay, as a type I TKI, crenolanib did not induce novel *FLT3* secondary TKD mutations, and only two crenolanib-treated patients demonstrate gatekeeper mutations in our study cohort. In addition, a novel *FLT3* extracellular mutation at K429E was detected in one patient with high VAF, which showed increased crenolanib $IC_{50}$. The structural basis for the drug resistance of *FLT3* K429E requires further investigation.

Another drug resistance mechanism of TKIs is expansion or activation of alternative signaling pathways[33,35,46–48]. Comparing the mutation profiles of chronic myeloid leukemia (CML) and AML patients may provide clues as to the mechanisms allowing BCR-ABL TKIs to demonstrate a low incidence of relapse and a long relapse-free latency. *BCR-ABL* was so far identified as the exclusive driver mutation in CML. Recent studies have not identified other recurrent mutations in a majority of CML patients by exome sequencing[49,50]. However, AML is a highly heterogeneous group of diseases exhibiting diverse coexisting mutations with great potential to generate multiple subclones during treatment[1,2,51,52]. In contrast to the mutual exclusivity of *FLT3*-ITD and D835 mutations seen in de novo AML (TCGA dataset), 35% of patients in TKI-naive postchemotherapy group and 51.4% of pre-TKI patients in the current cohort harbored both *FLT3*-ITD and D835 mutations[26], suggesting that both chemotherapy and type II FLT3 inhibitor treatment could select drug-resistant *FLT3* TKD clones. Furthermore, pre-TKI-treated crenolanib refractory and resistant patients exhibited more coexisting driver mutations compared to TKI-naive and crenolanib responders, respectively, suggesting a drug-resistant clone selection or induction pressure of TKI, including crenolanib treatment. In addition, previous studies have shown that *FLT3*-ITD demonstrated mutual exclusivity with *NRAS*, *KRAS*, *RUNX1*, *SF3B1*, *TP53*, *SRSF2* and *ASXL1* mutation in de novo AML[51], whereas in our current cohort, we observed high frequencies of coexisting *NRAS*, *KRAS*, *RUNX1*, *SF3B1*, *TP53*, *SRSF2* and *ASXL1* mutations with *FLT3* mutations respectively, suggesting a drug-resistant clonal selection or induction pressure of previous chemotherapy and/or previous TKI treatment.

Deciphering whether the drug treatment leads to the acquisition of resistance mutations, or whether resistance is indicative of the expansion of pre-existing sub-clones under selective pressure, is important to understanding drug resistance mechanisms that

may inform the design of better treatment regimens. New somatic mutations were detected in 10/30 samples with available paired samples, including mutations in *NRAS* ($n = 1$), *PTPN11* ($n = 2$), *ABL1* ($n = 1$), *FLT3* F691L ($n = 1$), *ASXL1* ($n = 1$), *BCORL1* ($n = 1$), *CEBPA* ($n = 1$), *WT1* ($n = 1$) and *IDH1* ($n = 1$) (Fig. 6, dark red). Notably, we did not detect the *IDH1* mutation in B32 and *CEBPA* mutation in A10 before crenolanib treatment by exome sequencing. However, targeted gene panels detected the *IDH1* and the *CEBPA* mutation before crenolanib treatment. The rest of the *NRAS* mutations as well as the majority of the mutations of *IDH2*, *RUNX1* and *STAG2* were present before crenolanib treatment with low VAF, and they expanded independently of the *FLT3* mutations. In contrast, mutations in *TET2*, *U2AF*1, *SF3B1* and *IDH1* were present in the same clone of *FLT3* mutations before crenolanib treatment and maintained during drug treatment. These data highlight the heterogeneity of AML clones and the expansion of pre-existing sub-clones under TKI selective pressure. Several alternative pathway mutation clones (Fig. 6, dark red) were not detected prior to treatment, which might be due to the limitation of current deep sequencing approaches to detect clones at very low levels. A higher prevalence of *NRAS* and *KRAS* mutations (26%) in de novo AML patients was detected in a recent study using deep targeted sequencing methods in comparison to 12% detected by WES in the TCGA AML cohort, indicating RAS mutations are frequently present in minor clones in de novo AML[51]. However, prolonged drug could also induce alternative pathway mutations shown by previous studies[47,53,54] and two *PTPN11* mutations in the current study (A14 and B27). Nevertheless, future high coverage sequencing methods before and during treatment may be needed to uncover drug-resistant and low-level clones.

In some instances, previous studies have neglected to analyze the co-occurrence or subclonal patterns of coexisting mutations. It is expected that mutations with different oncogenic pathways co-occurring in a distinct clone from the *FLT3* mutations can escape drug treatment and expand to become dominant clones. However, when they occur in the same clone as the *FLT3* mutation, the capacity of the concomitant clone to confer drug resistance needs further validation. Determination of the mutation co-occurrence pattern is important for understanding drug resistance and developing alternative treatment strategies. By analyzing the VAF dynamics, we could distinguish between co-occurrence and independent sub-clone patterns. Consistent with previous studies, we observed that *TET2* mutations co-occurred with *FLT3* mutations and induced primary crenolanib resistance. We also identified two patients with TP53 pathway mutations on the same clone with *FLT3* mutations. Previous studies have shown that the loss of TP53 led to midostaurin resistance.[34] In line with this study, we observed that knockout of TP53 conferred crenolanib resistance. In addition, we observed high co-occurrence of *IDH1* and *FLT3* mutations in crenolanib poor responders. Although we did not observe significant reduced crenolanib sensitivity of double mutation comparing to single mutation in ex vivo assays, this may be due to the models we used not accurately reflecting the patient actual biological condition. Furthermore, biologically relevant models are needed to assess the drug sensitivity of the combination mutations. In contrast, *IDH2*, *BCOR*, *STAG2* and signaling pathway mutations (*NRAS* and *PTPN11*) were present in subclones independent of *FLT3* mutations and escaped during crenolanib treatment, resulting in disease relapse. However, whether these mutations cause disease relapse alone or whether they cooperate with additional uncharacterized gene mutations and how exactly these mutations enable bypass of crenolanib inhibition are as-yet unknown.

Two major limitations of the current study must be noted. First, the use of VAF to analyze clonal architecture might be misinterpreted in cases where there is loss of heterozygosity and

presence of biallelic mutations. More sensitive and accurate sequencing methods, e.g. single cell sequence, allele-specific PCR or digital PCR, are needed to validate the clonal architectural data. Second, while AML is a highly heterogeneous disease with various cytogenetic and genetic alterations, the sample size of the current study is small. Future larger cohort studies are needed to validate and identify more recurrent coexisting mutations that confer crenolanib resistance.

Overall, we identify three distinct patterns of mutation dynamics during crenolanib treatment. Each clonal evolutionary pattern represents a distinct prognosis and indicates different potential strategies to circumvent drug resistance. Our study also provides clinical implications: comprehensive sequencing should be carried out on patient samples at the start and during the treatment in order to identify and pre-emptively target problematic clones. Additionally, while single-agent therapy with FLT3 inhibitors may be of marginal clinical benefit to patients with high VAF of *FLT3* mutations, it is imperative to combine a FLT3 inhibitor with chemotherapy or agents targeting cooperative lesions to achieve deep and durable remission.

## Methods

**Patients and samples**. Samples were obtained with informed consent and according to the Declaration of Helsinki under the institutional review board-approved protocols from patients who were enrolled on Phase II clinical trials of crenolanib (NCT 01522469 and NCT 01657682) in relapsed or refractory AML at the University of Texas Southwestern Medical Center and MD Anderson Cancer Center. Details of the clinical trials and results are summarized elsewhere.

**Assessment of outcomes**. Complete response was adapted according to the International Working Group Criteria for AML and key criteria were defined as follows: CR was defined as ≤5% blast in the borrow marrow, no circulating blasts, with complete blood count recovery (neutrophil count ≥1000/μL and platelet count ≥100,000/μL). CRi required the same criteria as CR with incomplete count recovery (neutrophil count <1000/μL and platelet count <100,000/μL). PR required all the hematologic values of CR with a decrease of ≥50% bone marrow blast but still >5%. HI was defined as ≥50% blast reduction in peripheral blood or bone marrow.

**Whole exome sequencing**. WES was performed as previously described[55] and summarized in a reporting summary statement as a Supplementary Information file. Briefly, DNA was extracted from leukemia patient specimens using Qiagen DNeasy according to the manufacturer protocols. For exome sequencing we used the Illumina Nextera capture probes and protocol (12 samples per capture group with each sample run on 3, 5 or 6 lanes) with libraries run on a HiSeq 2500 using paired-end 100 cycle protocols. Initial data processing and alignments were performed using our in-house workflows that we describe here briefly. For each flowcell and each sample, the FASTQ files were aggregated into single files for reads 1 and 2. BWA MEM version 0.7.10-r789[56] was used to align the read pairs for each sample-lane FASTQ file. As part of this process, the flowcell and lane information was kept as part of the read group of the resulting SAM file. The Genome Analysis Toolkit (v3.3) and the bundled Picard (v1.120.1579) were used[57] for alignment post processing. The files contained within the Broad's bundle 2.8 were used including their version of the build 37 human genome. The following steps were performed per sample-lane SAM file: (1) sorting and conversion to BAM via SortSam; (2) MarkDuplicates was run, marking both lane level standard and optical duplicates; (3) Read realignment around indels from the reads RealignerTargetCreator/IndelRealigner; (4) Base Quality Score Recalibration. The resulting BAM files were then aggregated by sample and an additional round of MarkDuplicates and indel realignment was carried out at the sample level. For genotyping, single-nucleotide variations (SNVs) and small indels were called using the UnifiedGenotyper and VarScan2[58]. Additionally, SNVs were called by MuTect[59]. Each VCF file was annotated using the Variant Effect Predictor v83[60] against GRCh37.

**Variant calling**. Since no paired normal tissue controls were available, we compiled a list genes associated with human hematologic cancers according to these two papers[61,62]. In total, 170 genes were selected (Supplementary Table 2). We used the global filtering as previously described[63]. On top of that, the following filters were used: (1) excluding variants found in more than 0.1% of ExAC samples; and excluding variants found in normal samples from more than 20% Beat AML normal controls; (2) including variant types: missense; frameshift; stop gain/loss; inframe insertion/deletion; protein altering; and tandem duplication for 127 genes list in Supplementary Table 1 (regular black font); (3) in addition, only frameshift, stop gain/loss and inframe insertion/deletion variants are considered for the following 43 genes (bold red font).

**Validation sequencing**. DNA extraction was performed the same as the 'WES' procedure. Libraries were created and hybridized using custom designed Nimblegen (SeqCap EZ) probes covering genes of interest containing both known variants and novel recurrent variants seen in other AML samples sequenced in-house. Because of the smaller library size, we ran 12 samples per lane. Paired-end 100 base reads were generated, aligned and post-processed using the WES protocol described above. For each unique variant position observed in the WES data, the number of reads supporting each observed allele in the validation library was determined using bam-readcount[56]. A variant was considered to be 'validated' by sequencing if there were at least three reads supporting the called variant.

**FLT3-ITD quantification with PCR**. *FLT3* exons 14 through 20 were amplified using forward primer 5'-GCAATTTAGGTATGAAAGCCAGC-3' and reverse primer 5'-CTTTCAGCATTTTGACGGCAACC-3'. The PCR products were then run on a 2% agarose gel stained with ethidium bromide and visualized and imaged under a Lummi imager (Roche Applied Science). The intensity of *FLT3*-ITD band and WT band were quantified by Image Lab software. The *FLT3*-ITD VAF was determined by calculating the ratio of *FLT3*-ITD to *FLT3*-ITD plus *FLT3* WT band intensity.

**MiSeq**. Exon 17 and exon 20 of *FLT3* were sequenced on serials samples from 20 patients to an average of 224,693 reads by MiSeq Next-Generation Sequencing (Illumina) as previously described[21,37] MiSeq data base calling accuracy was measured by the Phred quality score (Q score, https://www.illumina.com/documents/products/technotes/technote_Q-Scores.pdf). A Q score > 30 was used allowing for a mutation calling threshold to be set at 0.1%. *FLT3* exon 17 was amplified from genomic DNA (gDNA) using forward primer 5'- TCCCCAAGTC AGCAGAGAAC-3' and reverse primer 5'-GTTGCAGGACCCACAGACTT-3'. *FLT3* exon 20 was amplified using forward primer 5'- TTCCATCACCGGTAC CTCCTA -3' and reverse primer 5'-CCTGAAGCTGCAGAAAAACC -3'.

**Cell lines and reagents**. HEK 293T/17 cells (provided by Dr. Richard Van Etten) were maintained in Dulbecco's modified Eagle's medium (Invitrogen). Ba/F3 cells (ATCC) were maintained in RPMI 1640 (Invitrogen) supplemented with 15% WEHI-conditioned medium. Molm13, Molm14 and MV4–11 cells (DSMZ) were maintained in RPMI 1640. All mediums were supplemented with 10–20% fetal bovine serum (Atlanta Biologicals), L-glutamine, penicillin/streptomycin (Invitrogen) and fungizone (Fisher). Mycoplasma contamination was routinely tested (once per month). Only mycoplasma-free cells were used in the experiments. The cell lines were authenticated by internal FLT3-ITD PCR and small inhibitor screening as well as short tandem repeat analysis.

**Retroviral vector production and transduction**. *PTPN11* and *IDH1* mutations were generated using the QuikChange II XL site-directed mutagenesis kit (Agilent Technologies) on the respective pENTR vectors (GeneCopoeia GC-Z2134 and Invitrogen clone IOH11942) and cloned into a gateway compatible MSCV-IRES-puromycin retroviral vector or a Tet-inducible lentiviral vector, pInducer20 (Addgene, #444012) via Gateway Cloning System (Invitrogen). Retrovirus was produced by transfecting HEK 293T/17 cells together with an EcoPac helper plasmid). Lentivirus was produced by transfecting HEK 293T/17 cells together with psPAX2 (psPAX2 was a gift from Didier Trono (Addgene plasmid # 12260) and pLP/VSVG (Invitrogen). After 2 days, the virus containing supernatants were filtered, and infected to cells followed by flow cytometry (fluorescence-activated cell sorting (FACS)) sorting or puromycin selection.

**Ba/F3 IL-3 withdrawal assay**. Stably transduced Ba/F3 cells ($1 \times 10^6$) were washed three times and cultured in cytokine-free media. Viable cell number was determined on a Guava Personal Cell Analysis System (Millipore) every 1–2 days.

**Colony-forming unit (CFU) assay**. Bone marrow cells were harvested from BALB/c mice (The Jackson Lab, #000651), $Flt3^{ITD}$ and $Flt3^{ITD};Tet2^{+/-}$ transgenic BALB/c mice[33] (a kind gift from Evan Lind's lab, Oregon Health & Science University, Portland, OR). All mouse work was performed with approval from the Oregon Health & Science University Institutional Animal Care and Use Committee. For the BM transduction experiment, BM lineage-negative cells were enriched using Lineage Cell Depletion Kit (#130–090–858, Miltenyi Biotec), cultured overnight in medium containing IL-3, IL-6, and stem cell factor, and infected with retrovirus expressing FLT3 or IDH1 WT and mutants. Cytokines are purchased from PepreTech. A total of 2000 lineage-negative cells per well were seeded into 6-well plate with 1.1 mL of Methocult M3534 methylcellulose medium (StemCell Technologies) for 10 days. Images were taken and colonies (>50 cells) were counted via STEMvision™ colony counting software (StemCell Technologies). Colony cells were harvested, stained with cell surface markers and analyzed by FACS.

**FACS**. Cells were stained with antibodies (Biolegend) for 20 min at room temperature and washed twice with PBS. Membrane expression of CD11b, Sca1, c-KIT, Ter119 and CD71 were analyzed by FACS. All antibodies were used at 1:1000 dilution.

**Inhibitor assay**. Transformed Ba/F3 cells were seeded in 384-well plates (1250 cells per well) and exposed to increasing concentrations of crenolanib, trametinib or two-drug combination for 72 h. Cell viability was measured using a methanethiosulfonate (MTS)-based assay (CellTiter96 Aqueous One Solution; Promega), and read at 490 nm after 1–3 h using a BioTek Synergy 2 plate reader (BioTek). Cell viability was determined by comparing the absorbance of drug-treated cells to that of untreated controls (four replicates for each condition) set at 100%. The IC$_{50}$ values were calculated by a regression curve fit analysis using GraphPad Prism software.

**Evaluation of combinatorial effect of combination drugs**. We used Excess over Bliss (EOB) independence model[64] to quantify the synergy for crenolanib/trametinib combination at each drug concentration. EOB evaluates if the combined effect of two compounds is significantly greater or smaller than the combination of their individual (independent) effects and is measured by calculating the difference between the observed and predicted inhibition of the drug combination. For two single compounds with inhibition effects A and B, the predicted inhibition for the drug combination is calculated as $C = A + B - A \times B$. The two-agent combination inhibition is defined as AB. EOB can be calculated by $Z = AB - C$. Z Plus score (>0) indicates a synergistic effect, and Z minus score (<0) indicates an antagonistic effect. We define EOB Z-score ≥0.2 as strongly synergistic and ≤ −0.2 as strongly antagonistic. The predicted combination viability of drug A and B combination is defined as $(A + B - A \times B)$ %. Highest single-agent (HSA) model[64] was also used to evaluate if the combined effect of two drugs compounds is significantly greater or smaller than the higher individual drug effect.

**CRISPR targeting TP53**. A set of two single-guide RNA (sgRNAs) targeting *TP53* (sgRNA1 targeting *TP53* (TP53_1): 5'-GAGCGCTGCTCAGATAGCGA-3'; sgRNA2 targeting *TP53* (TP53_2): 5'-CCCCGGACGATATTGAACAA -3') as well as non-specific targeting control (NS: 5'-GGAGATATCAATCCTCCCGC-3')' were subcloned into plentiCRISPR v2 (a gift from Feng Zhang (Addgene plasmid # 52961)[65] according to the investigator's provided instructions. Lentiviruses were produced in HEK293T/17 cells using lentiCRISPR v2 coding for the respective sgRNA, Cas9 and puromycin resistance genes, and packaging plasmids psPAX2 and pLP/VSVG. Media containing viruses were spinoculated into Molm13 cells for 2 h at 35 °C, 2400 × g. Transduced cells were selected for puromycin resistance for 5 days and analyzed for the presence of genomic deletions using EndoT7 assay (not shown) and western blot analyses for TP53 protein (Cell Signaling, #2524, used at 1:1000 dilution) and glyceraldehyde 3-phosphate dehydrogenase (GAPDH; Thermo Fisher, #AM4300, used at 1:5000 dilution) loading control.

**Statistical analysis**. Statistical analyses were performed on GraphPad Prism software. The data were expressed as the mean ± SEM. Statistical significance was determined using two-tailed nonparametric Student's *t*-tests (Mann–Whitney test) or one-way analysis of variance (ANOVA) and expressed as *p* values (*$p < 0.05$, **$p < 0.01$, ***$p < 0.001$ and ****$p < 0.0001$).

**Reporting summary**. Further information on experimental design is available in the Nature Research Reporting Summary linked to this article.

## Data availability

All sequence data have been deposited at dbGaP and Genomic Data Commons. The study ID is 29125 and the accession number is phs001628. In addition, all data can be accessed and queried through our online, interactive user interface, Vizome, at www.vizome.org.

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

## Acknowledgements

We thank Judy Ho, June Lam and Ting-Chun Yeh for organizing the patient sample information and discussion. We acknowledge Jaime Faulkner and Matthew Newman for harvesting the mouse bone marrow cells. We thank Dorian LaTocha and Brianna Garcia for help in the FACS sorting; and Kara Johnson for general help. This work was supported in part by The Leukemia & Lymphoma Society Beat AML Program, the V Foundation for Cancer Research, the Gabrielle's Angel Foundation for Cancer Research and the National Cancer Institute (1R01CA183947–01; 1U01CA217862–01; 1U54CA224019-01; 3P30CA069533-18S5). H.Z. received a Collins Medical Trust research grant. S.D.B. was supported by the National Cancer Institute (5R01CA138744-08).

## Author contributions

S.S. and H.Z. extracted gDNA from patient samples and organized the patient sample information. H.Z. and A.R.S. performed the experiments. G.B., G.G.-M., F.R., H.M.K., J.C. and R.C. conducted the clinical trial, coordinated and managed the collection of samples. V.J., H.H., Y.L.L., R.S. and C.F. organized the patient sample information and participated in data analysis and discussion. T.N. performed the CRISPR-Cas9 *TP53* knockout experiment. A.C., R.H., C.L. and R.S. performed the whole exome and validation sequencing. D.B., L.W., E.S., B.W. and S.K.M. carried out statistical and bioinformatic analyses of genetic and clinical data. B.W. conceived of the Vizome platform, provided oversight for development of the platform and helped provide project oversight for experimental design, workflow development, data processing, management, analysis and dissemination and method development. D.B. assisted with NGS pipeline and data management. L.W. wrote all of the software for the Vizome platform, developed the platform as well as novel visualizations for data integration and display. E.S. helped with the exome data processing, data management and developed the data dissemination workflows. C.E. participated in analyzing and graphing the data. E.L. and B.J.D. provided important research materials. D.R.B. and S.D.B. performed the Miseq sequence analysis. H.Z. and J.W.T. analyzed/interpreted the data and wrote the manuscript. All authors contributed to the final manuscript.

## Additional information

**Competing interests:** Research support for J.W.T. is received from Aptose, Array, AstraZeneca, Constellation, Genentech, Gilead, Incyte, Janssen, Seattle Genetics, Syros, Takeda; co-founder of Vivid Biosciences. B.J.D. potential competing interests-- Scientific Advisory Board: Aileron Therapeutics, ALLCRON, Cepheid, Gilead Sciences, Vivid Biosciences, Celgene & Baxalta (inactive); SAB & Stock: Aptose Biosciences, Blueprint Medicines, Beta Cat, GRAIL, Third Coast Therapeutics, CTI BioPharma (inactive); Scientific Founder & Stock: MolecularMD; Board of Directors & Stock: Amgen; Board of Directors: Burroughs Wellcome Fund, CureOne; Joint Steering Committee: Beat AML LLS; Clinical Trial Funding: Novartis, Bristol-Myers Squibb, Pfizer; Royalties from Patent 6958335 (Novartis exclusive license) and OHSU and Dana-Farber Cancer Institute (one Merck exclusive license) H.H., Y.L.L., R.S., C.F. and V.J. are employees of AROG Pharmaceuticals. The remaining authors declare no competing interests.

