## [Peer Review File · Nature Communications]

Reviewers' comments:

Reviewer #1 (Remarks to the Author):

Zhang et al. studied the mechanisms underlying the clinical resistance to the pan-FLT3 inhibitor Crenolanib. Although the findings are very interesting and are potentially extremely useful due to their applicability in clinics, there are aspects that need to be further studied.

- In general, there are essential negative and positive controls missing. All the experiments run with transduced cell lines need to be compared against an empty vector-transduced cell line.

Additionally, single mutation-transduced cells are also necessary.

- Authors used in this paper mainly 4 different types of specimens: primary human AML samples, murine pro-B cell line (Ba/F3), mouse lineage-depleted hematopoietic cells, human AML cell line (Molm-14). Although the rationale to use such diverse biological samples is acceptable, conclusions and data analysis became weaker if the most important data is not generated in human AML cells.

- There are some stats missing in Figures.

- Synergism is claimed although no mathematical method is being described that support synergism between treatments. It is necessary to calculate the amount of synergism (if any) using a mathematical method.

Reviewer #2 (Remarks to the Author):

The manuscript by Zhang et al describes analysis of AML patients treated with the FLT3 inhibitor, crenolanib. The authors perform whole exome sequencing on a large cohort of patients before and after crenolanib treatment to evaluate the mutational profile and how it changes following therapy. The results in a number of possible outcomes including mutations of NRAS and IDH2 arising as FLT3-independent subclones, and TET2 and IDH1 co-occurring with FLT3 mutant clones. Collectively, the data suggest diverse genetic/epigenetic mechanisms of crenolanib resistance.

The strengths of this manuscript are the novelty of using WES for a crenolanib study, serial analysis of a large cohort of patients, generation of what is likely a valuable dataset, and the significance of better understanding mechanisms of FLT3 resistance in AML. Unfortunately, overall enthusiasm is tempered by two broad concerns:

1) Generally speaking, the study is a mile wide and an inch deep. As such, there is no compelling conclusion or take home message that emerges from the overall study. The authors comprehensively catalog the changes that occur following crenolanib therapy, but do very little to investigate downstream mechanisms related to any particular mutations. As such, the work is very descriptive. There are multiple potentially exciting and important lines of investigation to pursue. The manuscript would greatly benefit from further developing these stories (likely several papers could arise from the foundation created by the initial work).

2) The data presentation in both the text and figures is often confusing and quite difficult to follow. This is partially due to the breadth of the findings, but it is nonetheless a challenge to sort through any part of the data and fully understand the implications/conclusions.

Reviewer #3 (Remarks to the Author):

The manuscript by Zhang et al. is an interesting examination of possible modes of resistance of FLT3 mutation positive patients receiving crenolanib TKI therapy. 59 patients are studied, some previously exposed to TKI, some naive. The main findings are 1) resistance was not associated with FLT3 activation loop mutations, 2) a different spectrum of mutations in patients previously exposed to TKI as opposed to TKI naive patients, 3) sequencing data suggested independent clonal evolution of N-ras mutation cases, while FLT3 mutation cases with TET2/IDH appeared to persist with crenolanib therapy.

There are several issues that should be addressed, which will be highlighted here in the order of appearance of the manuscript.

1. Line 98. It is not clear if the post-therapy samples were taken at morphological or hematological relapse, and what the median and range of the blast counts were that were subsequently sequenced. This obviously has an impact on the VAF calculation and interpretation.

2. Line 147. The difference in the mutational landscape between these groups is important, and "a trend" is not adequate for a major point of the paper. There needs to be a statistical analysis reported of the distribution, with CI. The readers can infer from the strength of the relationship.

3. Line 155. It is worrisome that the VAF of these are all around 50%-this makes sense if all mutations were heterozygous in 100% blasts. This also pertains to the results shown in line 164, and the N ras VAF results. The potential issue is that several centers have observed that the performance of sequencing panels varies from loci to loci, and one cannot assume that each loci and gene VAF

results are similarly robust. Thus, have the authors done spike in experiments for N ras, TET2, DMT3A, etc, to see if the VAF is accurate across a broad range of inputs? If not, the data is very difficult to interpret.

4. Sections beginning lines 127, 144, and 177. Do the authors have RNA seq data to correlate mutations with gene expression? This could be especially powerful in typing to their drug sensitivity data (presented later).

5. Line 189. Single cell genotyping in AML and other diseases has suggested that this assumption may not be solid.

6. General. The paper has many interesting and noteworthy observations. Many, however, are based on very small numbers of patients. This is unavoidable, and OK, but in the Discussion, the results and interpretations should perhaps be sorted based on the the likely "wobble" of the data given the above.

7. General. Could the authors go back to the cases where resistance is associated with a new mutation (e.g., N ras) and perform more sensitive analysis of the pre-cre samples to try to find the mutation? (allele specific PCR, digital PCR, etc)?

We appreciate the detailed reviews and insightful suggestions from the reviewers. We have addressed or performed further experiments and analysis to answer all the concerns and questions as summarized in the following point by point response.

Reviewers' comments:

Reviewer #1 (Remarks to the Author):

Zhang et al. studied the mechanisms underlying the clinical resistance to the pan-FLT3 inhibitor Crenolanib. Although the findings are very interesting and are potentially extremely useful due to their applicability in clinics, there are aspects that need to be further studied.

- In general, there are essential negative and positive controls missing. All the experiments run with transduced cell lines need to be compared against an empty vector-transduced cell line. Additionally, single mutation-transduced cells are also necessary.

Answer: We agree that empty vector controls should be used, and we have added empty controls to Fig. 1b and 1c. In addition, we transduced FLT3 D835Y cells with an empty vector. FLT3 D835Y or FLT3 D835Y/empty vector demonstrated similar crenolanib sensitivities.

In addition, in all instances where we are testing the impact of a second mutant gene (in addition to mutant FLT3) on crenolanib response, we also include the wild type versions of each gene (also co-expressed with mutant FLT3) as controls. For example, we have used PTPN11 WT/FLT3 D835Y as a control for PTPN11 A72D/FLT3 D835Y; IDH1 WT/FLT3 D835Y as a control for IDH1 R132H /FLT3 D835Y.

Figure legend: Graph depicts mean \pm SEM of cell viability of Ba/F3 cells expressing FLT3 D835Y or FLT3 D835Y in combination with an empty vector treated with dose gradients of crenolanib for 72h determined by MTS assay as described in Materials and Methods.

- Authors used in this paper mainly 4 different types of specimens: primary human AML samples, murine pro-B cell line (Ba/F3), mouse lineage-depleted hematopoietic cells, human AML cell line (Molm-14). Although the rationale to use such diverse biological samples is acceptable, conclusions and data analysis became weaker if the most important data is not generated in human AML cells.

Answer: we agree that primary human AML cells are the most suitable model to use, and fortunately all of the sequencing in this manuscript was performed on tissue from AML patients. Unfortunately, since this is a retrospective study, further material from these patients is no longer available due to the poor prognosis of these patients. In addition, the use of primary cells from other AML patients is hindered by the fact that de novo AML samples rarely carry FLT3+NRAS, FLT3+PTPN11 or FLT3+TP53 co-existing mutations—in contrast to what we observed recurrently in the relapsed AML patients on this study. Therefore, we were forced to use cell line models of these double mutations seen in crenolanib resistant patients to characterize the crenolanib sensitivity of single versus co-occurring mutations. We anticipate future validation studies with fresh or viably frozen primary samples as a follow-up to this study.

There are some stats missing in Figures.

- Synergism is claimed although no mathematical method is being described that support synergism between treatments. It is necessary to calculate the amount of synergism (if any) using a mathematical method.

We agree. We have added synergism calculation in the supplementary data. Briefly, we have used Excess over Bliss (EOB)¹ independence model to quantify the synergy for crenolanib/trametinib combination at each drug concentration. We also used Highest single agent (HSA) model¹ to evaluate if the combined effect of two drugs compounds is significantly greater or smaller than the higher individual drug effect. Detailed information can be found in Supplementary materials and methods, and Supplementary Fig. 4 and Fig. 6.

Supplementary Fig. 4

b

Drug concentration (nM)	1000.0	333.3	111.1	37.0	12.3	4.1	1.4	0.5	0
EOB Z score (FLT3 D835Y)	0.006	0.003	0.002	0.009	0.373	0.478	0.311*	0.267*	0
EOB Z score (FLT3 D835Y/ PTPN11 WT)	-0.017	-0.058	-0.016	-0.004	0.216	-0.122	-0.048	-0.011	0
EOB Z score (FLT3 D835Y/ PTPN11 A72D)	-0.085	-0.070	-0.005	0.045	0.372	0.321	0.199	0.275	0

Figure legend: Crenolanib and trametinib combination therapy demonstrate synergistic toxicity. **a**, Graph depicts mean \pm SEM of cell viabilities of Ba/F3 cells expressing FLT3 D835Y along or with PTPN11 WT or PTPN11 72D compound mutation treated with crenolanib, trametinib, or two drug equal concentration combination. The predicted viability of crenolanib and trametinib combination is shown in blue according to EOB calculation as defined in the materials and methods. **b**, The table summarizes the EOB z score of crenolanib/trametinib combination at each concentration. * indicates that the drug combination effect is smaller than the HSA effect. Red highlights that crenolanib+trametinib combination demonstrates strongly synergistic effect.

Supplementary Fig. 6

Figure legend: Crenolanib and trametinib combination therapy demonstrate synergistic toxicity. **a**, Graph depicts mean \pm SEM of cell viabilities of Ba/F3 cells expressing FLT3 D835Y along or with PTPN11 WT or PTPN11 72D compound mutation treated with crenolanib, trametinib, or two drug equal concentration combination. The predicted viability of crenolanib and trametinib combination is shown in blue according to EOB calculation as defined in the materials and methods. **b**, The table summarizes the EOB z score of crenolanib/trametinib combination at each concentration. * indicates that the drug combination effect is smaller than the HSA effect. Red highlights that crenolanib+trametinib combination demonstrates strongly synergistic effect.

Reviewer #2 (Remarks to the Author):

The manuscript by Zhang et al describes analysis of AML patients treated with the FLT3 inhibitor, crenolanib. The authors perform whole exome sequencing on a large cohort of patients before and after crenolanib treatment to evaluate the mutational profile and how it changes following therapy. The results in a number of possible outcomes including mutations of NRAS and IDH2 arising as FLT3-independent subclones, and TET2 and IDH1 co-occurring with FLT3 mutant clones. Collectively, the data suggest diverse genetic/epigenetic mechanisms of crenolanib resistance.

The strengths of this manuscript are the novelty of using WES for a crenolanib study, serial analysis of a large cohort of patients, generation of what is likely a valuable dataset, and the significance of better understanding mechanisms of FLT3 resistance in AML. Unfortunately, overall enthusiasm is tempered by two broad concerns:

1) Generally speaking, the study is a mile wide and an inch deep. As such, there is no compelling conclusion or take home message that emerges from the overall study. The authors comprehensively catalog the changes that occur following crenolanib therapy, but do very little to investigate downstream mechanisms related to any particular mutations. As such, the work is very descriptive. There are multiple potentially exciting and important lines of investigation to pursue. The manuscript would greatly benefit from further developing these stories (likely several papers could arise from the foundation created by the initial work).

We agree that this study describes the landscape of mutational profiles that confer crenolanib resistance (co-occurring RAS-pathway, TET2, TP53, etc. mutations). While we have provided some mechanistic validations within the space permitted in a single article, the reviewer is absolutely correct that in depth mechanistic characterizations of FLT3 mutation co-occurring mutations will need to be performed in future studies. Notably, certain of the observations in this study, such as crenolanib resistance driven by outgrown subclones with RAS pathway mutations are somewhat self-explanatory, since crenolanib is a FLT3 inhibitor without targeting capacity for RAS pathway kinases, however, other events will require more detailed investigations.

2) The data presentation in both the text and figures is often confusing and quite difficult to follow. This is partially due to the breadth of the findings, but it is nonetheless a challenge to sort through any part of the data and fully understand the implications/conclusions.

We agree that this dataset covers a lot of ground, which results in a narrative that can be difficult to follow. We have made revisions to the text to pull together the manuscript into a more cohesive unit that should be easier for readers to navigate.

We have modified the end part of the introduction, to give a brief idea what questions we were asking and the manner and order we were going to address those questions. We have also added an introduction at the beginning of each result part to help understand the rationale and the aim of that particular result part. Briefly, our study aimed to answer what are the

potential mechanisms associated with crenolanib resistance from three major aspects: FLT3 secondary mutations; FLT3 co-occurring mutation; and FLT3 mutation independent subclone mutations.

Reviewer #3 (Remarks to the Author):

The manuscript by Zhang et al. is an interesting examination of possible modes of resistance of FLT3 mutation positive patients receiving crenolanib TKI therapy. 59 patients are studied, some previously exposed to TKI, some naive. The main findings are 1) resistance was not associated with FLT3 activation loop mutations, 2) a different spectrum of mutations in patients previously exposed to TKI as opposed to TKI naive patients, 3) sequencing data suggested independent clonal evolution of N-ras mutation cases, while FLT3 mutation cases with TET2/IDH appeared to persist with crenolanib therapy.

There are several issues that should be addressed, which will be highlighted here in the order of appearance of the manuscript.

1. Line 98. It is not clear if the post-therapy samples were taken at morphological or hematological relapse, and what the median and range of the blast counts were that were subsequently sequenced. This obviously has an impact on the VAF calculation and interpretation.

Both morphological and hematological relapses were taken into account. When patients' bone marrow specimens are available, relapse was evaluated according to the bone marrow morphology; whereas when bone marrow samples were not available, hematological relapse was evaluated. We have included detailed information on these data points for each patient, which can be found in Supplementary Table 1_Summary of all patients' information, sample type tab, and these data will be helpful in interpretation of VAF data, as the reviewer suggests.

2. Line 147. The difference in the mutational landscape between these groups is important, and "a trend" is not adequate for a major point of the paper. There needs to be a statistical analysis reported of the distribution, with CI. The readers can infer from the strength of the relationship.

Answer: we agree. We have provided this information in the supplementary Fig.1 and supplementary Fig.2. While we agree that it is preferable to discuss data points that achieve a certain degree of statistical significance, we would submit that we are researching a very heterogeneous population, which renders the sample size and statistical power for many genetic subsets too small to achieve the most rigorous of statistical thresholds. We feel that it is still appropriate to highlight certain "trends" that have not achieved statistical significance, but which merit further exploration, especially for those events where we have provided additional follow-up data from wet lab experimentation.

Figure legend: **a**, Graph depicts log-transformed mean and 95% confidence intervals of odd ratios of gene mutation frequencies of pre-TKI group comparing to TKI naïve group. **b**, Graph depicts log-transformed mean and 95% confidence intervals of odd ratios of gene mutation frequencies of crenolanib non-responders comparing to crenolanib responders.

3. Line 155. It is worrisome that the VAF of these are all around 50%-this makes sense if all mutations were heterozygous in 100% blasts. This also pertains to the results shown in line 164, and the N ras VAF results. The potential issue is that several centers have observed that the performance of sequencing panels varies from loci to loci, and one cannot assume that each loci and gene VAF results are similarly robust. Thus, have the authors done spike in experiments for N ras, TET2, DMT3A, etc, to see if the VAF is accurate across a broad range of inputs? If not, the data is very difficult to interpret.

Answer: We agree that “spike in” is a good strategy to check the accuracy of sequencing across a range of inputs. Unfortunately, we were not aware of this when we started the sequencing several years ago and did not have a spike in in this study. Instead, based on the best practices at the time, we assessed coverage and performed targeted sequencing to validate the whole exome sequencing data. The exome sequencing coverage, is viewable at: www.vizome.org (user: reviewer; pwd: fqw2ldfv; gene model module). Overall consistent VAFs between whole exome sequencing and validation sequencing were observed, including NRAS, TET2, and DNMT3A. The consistent data obtained from these two methods, could to a certain degree provide some confidence to the interpretation of the results. As shown in the below figure, the VAFs detected by these two methods are closely correlated ($r=0.758$, $p<0.0001$).

Figure legend: Scatter plot depicts Pearson correlation of VAF of a specific variant in Supplementary Table 4 detected by exome sequencing and validation sequencing.

4. Sections beginning lines 127, 144, and 177. Do the authors have RNA seq data to correlate mutations with gene expression? This could be especially powerful in typing to their drug sensitivity data (presented later).

Answer: we attempted to isolate and analyze RNA from baseline and resistant samples. Unfortunately, only unprocessed BM or whole peripheral blood were initially collected and stored. We were not able to recover high quality and sufficient RNA from these samples for RNAseq analysis.

5. Line 189. Single cell genotyping in AML and other diseases has suggested that this assumption may not be solid.

We agree that single cell sequence is more accurate. We have mentioned this in the discussion study limitation part. We plan to perform single cell genotyping for future projects where fresh or viably frozen cells are available.

6. General. The paper has many interesting and noteworthy observations. Many, however, are based on very small numbers of patients. This is unavoidable, and OK, but in the Discussion, the results and interpretations should perhaps be sorted based on the the likely "wobble" of the data given the above.

We agree that, due to the small sample size and the heterogeneity of AML, future larger cohort validation is needed. We have emphasized the limitation of our study in the discussion. However, we do feel that our findings of diverse mechanisms of relapse and resistance reported here are still valid and impactful, since AML is a heterogeneous disease with various and dynamic cytogenetic and genetic alterations, and this study allows us to assess this longitudinally.

7. General. Could the authors go back to the cases where resistance is associated with a new mutation (e.g., N ras) and perform more sensitive analysis of the pre-cre samples to try to find the mutation? (allele specific PCR, digital PCR, etc)?

Thank you for your suggestion. It's a good experiment to determine whether the mutations are present before crenolanib in small subclones or emerged during crenolanib treatment. From literature, both instances have shown evidence.²⁻⁵ To address this question, we have done further analysis of our MiSeq and Exome sequencing data. We show instances where mutations were detectable by MiSeq at very low levels (less than 1%), however could not be detected by exome sequencing with a relatively low coverage (Example 1). We also show instances where mutations that were prevalent at relapse, however were not detected with coverage of ~60-100X by exome sequencing before the treatment, suggesting that those mutations were either

not present or were below the level of detection for this coverage – an important point for design of clinical sequencing applications (Example 2 B27 and B32). Perhaps most interestingly, however, we also see evidence from MiSeq of variants that were seen at extremely low abundance (~1-10 calls out of 100-300,000 reads) before crenolanib treatment (B06) and then expanded during crenolanib treatment. These low baseline calls occur quite frequently, and raise the question of whether calls at this level can be confidently interpreted as bona fide mutations versus sequencing artifacts. For our study, we only called mutations if they were present in the MiSeq data at greater than 0.1% frequency. Please see below for these examples, which we have now discussed in the manuscript.

Example 1: Low VAF of variants found by high coverage Miseq (in red font), but not by exome sequencing.

The below table summarized the detection of low VAF of FLT3 tyrosine kinase domain mutations by Miseq. However, the same variants were not detected by exome sequencing (mean ± standard error of the mean of exome sequencing coverage between amino acid 833 and 841: 169±35).

Table. Miseq analysis of FLT3 tyrosine kinase domain mutations.

Patient	Sample time course/day	Source of DNA	Baseline FLT3 status	A833S (GCT) T (A->S)	Coverage A833S	D835H (GAT) C (D->H)	Coverage D835H	D839Y (GAT) T (D->Y)	Coverage D839Y	D839G (GAT) G (D->G)	Coverage D839G	N841K (AAC) A (N->K)	Coverage N841K	R834D835I836 (CGAGATA TC) CGCCCC (RDI->RP)	Coverage R834D835I836->RP
A04	C1D1	serum	ITD	8	201582	2	201548	1945 (0.96%)	203442	7	199954	7	199954	0	195504
	C2D1	NA		na	dropped out	na	dropped out	na	dropped out	na	dropped out	na	dropped out	na	dropped out
	C4D1	NA		4	185833	14	185818	31	187380	12	182663	6	182663	0	179606
A05	C1D1	serum	835 and ITD	5949 (3.33%)	178598	2	176567	19	178153	6	174655	5	174655	0	169385
	C2D1	NA		11	279026	39	279001	27	280699	15	275706	12	275706	0	270653
A08	C1D1	serum	ITD/D835	7	161435	2	161402	483 (0.29%)	165034	435 (0.27%)	161689	6	161689	756 (0.48%)	162503
	C1D1	serum		9	140308	9	140305	36	141173	33	140007	8	140007	864 (0.65%)	133252
	C2D1 (C2D26)	BMA		10	166423	11	166404	13	167185	12	165389	7	165389	0	160706
	C3D1 (C2D26)	BMA		10	141608	17	141594	8	143459	11	140103	5	140103	0	138884
	C5D1			8	477283	54	477244	11	477803	9	476402	15	476402	1	468881
B21	C1D1	serum	ITD/D835	4	76676	158 (0.21%)	76675	9	77140	13	74859	311 (0.44%)	74859	0	70357
	C1D1	serum		10	157618	1089 (0.69%)	157602	12	157183	12	154267	17	154267	0	150955
	C2D1	BMA		8	158847	14	158823	11	158848	15	155074	7	155074	0	152446
	C3D1 (C2D26)	BMA		9	174660	293 (0.17%)	174620	14	174130	10	171036	9	171036	0	169139
	C5D1 (C4D28)	BMA		7	175004	333 (0.19%)	174989	12	174990	7	172601	6	172601	0	169827

Example 2: high coverages of a specific gene location were shown in both initial and during treatment samples, whereas variants of high VAF were only detected in during treatment samples, highly suggesting the acquisition of new variants. Below table shows three example: the first is Miseq date; the other two are from exome sequencing data.

Patient	Sample ID	Sample time	Source of DNA	PTPN11 A72D (G/A)	Coverage
B27	14-00885	C1D1	Blood	0	67
	14-00884	C2D1	BMA	0	65
	16-00958	C7D9	BMA	49 (35%)	139

Patient	Sample ID	Sample time	Source of DNA	IDH1 R132H (C/T)	Coverage
B32	16-00965	C1D1	Blood	0	76
	16-00966	C2D1	BMA	31 (38%)	82

Patient	Sample time	Source of DNA	Baseline FLT3 status	F691L (TTT) C (F->L)	F691L (TTT) G (F->L)	Coverage F691L (TTT) C (F->L)	Coverage F691L (TTT) G (F->L)
B06	C1D1	serum	D835 and	9	5	277343	277294
	C2D1	NA	ITD	2398 (1.6%)	86946 (36%)	149756	151969

References

1. Fouquier, J. & Guedj, M. Analysis of drug combinations: current methodological landscape. *Pharmacol. Res. Perspect.* **3**, e00149 (2015).
2. Piloto, O. *et al.* Prolonged exposure to FLT3 inhibitors leads to resistance via activation of parallel signaling pathways. *Blood* **109**, 1643–1652 (2007).
3. Metzeler, K. H. *et al.* Spectrum and prognostic relevance of driver gene mutations in acute myeloid leukemia. in *Blood* **128**, 686–698 (2016).
4. Irving, J. *et al.* Ras pathway mutations are prevalent in relapsed childhood acute lymphoblastic leukemia and confer sensitivity to MEK inhibition. *Blood* **124**, 3420–3430 (2014).
5. Farrar, J. E. *et al.* Genomic profiling of pediatric acute myeloid leukemia reveals a changing mutational landscape from disease diagnosis to relapse. *Cancer Res.* 1–10 (2016). doi:10.1158/0008-5472.CAN-15-1015

Reviewers' comments:

Reviewer #1 (Remarks to the Author):

Following the comments provided, the authors have included all controls regarding transduced cells. However, no stats are included in Figure 4C, 4D and 5B as originally requested.

Although desirable, obtaining primary AML cells can be challenging. However, as the paper focus on AML, the data provided using cell lines should include at least 2 different AML cell lines (in order to avoid any effect related to clonality). Authors used Molm-13 and Molm-14 which are sister cell lines. Most of the data that biologically support the main claims are obtained using a murine pro-B leukemia cell line (Ba/F3). This cell line is not even a myeloid leukemia. Cell context is important in all processes. The Flt3 signaling dependence of Ba/F3 cell line presents an attractive model to study the implication of this protein in drug resistance and pathogeneity. However, taking into account that Molm-13 does not express Flt3 (according to the repository ATCC), it will be a good opportunity to express Flt3 mutants and confirm the findings in the B cell line. Additionally, many AML cell lines have been sequenced and mutations in Flt3 and other genes are published.

Reviewer #2 (Remarks to the Author):

Despite significant concerns noted by reviewers, there is no significant change to the paper.

Reviewer #3 (Remarks to the Author):

The authors have satisfied my concerns.

Reviewer #1 (Remarks to the Author):

Following the comments provided, the authors have included all controls regarding transduced cells. However, no stats are included in Figure 4C, 4D and 5B as originally requested.

Answer: we apologize for missing this question. We have added statistical analyses to Figure 4C, 4D and 5B. Statistical significance was determined using a two-tailed nonparametric Student's t-test (Mann-Whitney test) comparing each group to non-treated group and expressed as * $p < .05$. Graphs shown below are Fig. 4c and Fig. 4d. No significant differences were observed in Figure 5B.

Although desirable, obtaining primary AML cells can be challenging. However, as the paper focus on AML, the data provided using cell lines should include at least 2 different AML cell lines (in order to avoid any effect related to clonality). Authors used Molm-13 and Molm-14 which are sister cell lines. Most of the data that biologically support the main claims are obtained using a murine pro-B leukemia cell line (Ba/F3). This cell line is not even a myeloid leukemia. Cell context is important in all processes. The Flt3 signaling dependence of Ba/F3 cell line presents an attractive model to study the implication of this protein in drug resistance and pathogeneity. However, taking into account that Molm-13 does not express Flt3 (according to the repository ATCC), it will be a good opportunity to express Flt3 mutants and confirm the findings in the B cell line. Additionally, many AML cell lines have been sequenced and mutations in Flt3 and other genes are published.

Answer: 1) We agree. We have expressed FLT3 D835Y and FLT3 K429E in mouse bone marrow stem cells and performed a crenolanib sensitivity assay. Consistent with the Ba/F3 system, reduced crenolanib sensitivity is observed in bone marrow cells expressing FLT3 K429E compared with cells expressing FLT3 D835Y. Please find these new data in Supplementary Fig. 4a.

Figure legend: Graph depicts mean \pm SEM of colony numbers (left panel) or fold change of colony number (normalized to the mean colony number of no treatment, right panel) for three replicates of mouse bone marrow cells transduced with retroviral vector expressing FLT3 D835Y or FLT3 K429E treated with gradient concentrations of crenolanib for 10 days. Statistical analysis were performed using one way ANOVA together with Dunn's multiple comparisons tests and expressed as: * $p < .05$ and ** $p < .01$.

2) We have also expressed PTPN11 WT or mutant in MV4-11 cells using a doxycycline inducible lentiviral system. We observed that PTPN11 A72D demonstrated reduced crenolanib sensitivity comparing to PTPN11 WT or empty vector control (without doxycycline.)

Figure legend: Representative graph depicts mean \pm SEM of cell viability of MV4-11 cells expressing a doxycycline (Dox) inducible PTPN11 WT or PTPN11 A72D vector treated with dose gradients of crenolanib in the presence or absence of Dox (1 μ g/ml) for 72h determined by MTS assay as described in Materials and Methods.

Reviewers' comments:

Reviewer #1 (Remarks to the Author):

The authors have not addressed the concerns raised during the first and second revisions. No significant improvement has been made regarding the lack of experimental data regarding the role of Flt3 in resistance to crenolanib in AML. Instead, authors have transduced mutated Flt3 isoforms (without comparing to the Flt3wt-transduced AML cells and empty-vector transduced cells) in murine bone marrow cells.

We appreciate the opportunity to address and provide data to support our manuscript. We have performed additional experiments and addressed reviewer 1's concern with regard to the FLT3 mutations driving crenolanib resistance. The following represent our specific comments and replies:

The authors have not addressed the concerns raised during the first and second revisions. No significant improvement has been made regarding the lack of experimental data regarding the role of Flt3 in resistance to crenolanib in AML.

We disagree and would point to substantial new data that we have added to the manuscript in response to reviewer points during both revisions as detailed here:

In this cohort, we identified three FLT3 secondary mutations (one patient with FLT3 K429E, two patients with gatekeeper F691 mutations) with persistent or increased allelic burdens during crenolanib treatment (Revision3 figure 1a). We have shown that K429E alone or in combination with FLT3 D835Y exhibits reduced crenolanib sensitivity (Revision3 figure 1b-d). We further confirmed the crenolanib resistance with transduced marine bone marrow cells (Revision3 figure 1e). The gatekeeper F691 mutations do not transform cells alone, so we have shown that FLT3 F691L/D835Y demonstrated decreased crenolanib resistance (Revision3 figure 1c-d). In addition, FLT3 gatekeeper mutations were previously shown to induce crenolanib resistance in other publications¹ and have been well characterized to cause drug resistance to other FLT3 inhibitors (sorafenib, quizartinib, etc.)^{2,3}. We have now used FLT3 WT and empty vector as negative controls (do not respond to crenolanib), and we have used FLT3 D835Y and FLT3-ITD transformed cells as positive controls (are sensitive to crenolanib). In this context, the FLT3 K429E variant exhibits significantly less crenolanib response than FLT3-ITD or FLT3-D835 variants. With these data, we feel that we have convincingly shown that FLT3 secondary mutations contribute to crenolanib resistance in a minority of this cohort (3 patients out of the cohort). In particular, this includes the novel variant, K429E, which has not previously been detected or characterized.

Figure legend: FLT3 K429E and FLT3 F691L demonstrates reduced crenolanib sensitivity. a, VAFs of non-D835 FLT3 mutations during crenolanib treatment. b-c, Ba/F3 cells expressing FLT3 K429E, FLT3 K429E/D835Y, FLT3 K429E/F691L demonstrate reduced crenolanib sensitivity. Graphs depict mean ± SEM of cell viabilities of Ba/F3 cells

expressing empty vector, FLT3 WT or mutants treated with dose gradients of crenolanib for 72h determined by MTS. **d**, Mean \pm SEM of crenolanib IC₅₀ and IC₉₀ values of Ba/F3 cells transformed with FLT3 WT and mutants as presented in panels b-c. Graphs and images shown are representatives from more than three experiments. Statistical significance was assessed using one way ANOVA and Kruskal-Wallis test comparing each condition to the respective FLT3 D835Y and expressed as: * p<.05; ** p<.01. **e**, Graph depicts mean \pm SEM of fold change of colony number (normalized to the mean colony number of no treatment) for three replicates of mouse bone marrow cells transduced with retroviral vector expressing empty vector, FLT3 WT or mutants treated with gradient concentrations of crenolanib for 10 days. Statistical analyses were performed using one way ANOVA together with Dunn's multiple comparisons tests comparing crenolanib treated cells to each individual untreated control and expressed as: * p<.05 and ** p<.01.

This figure is a modified blend of data from Figure 1 and supplementary Figure 1 from the manuscript, assembled for convenience of review.

Instead, authors have transduced mutated Flt3 isoforms (without comparing to the Flt3wt-transduced AML cells and empty-vector transduced cells) in murine bone marrow cells.

*The question being asked was whether FLT3 K429E is less sensitive to crenolanib than crenolanib-sensitive FLT3 variants (such as D835Y mutations). Hence, the appropriate way to address this question is to elicit colony formation from FLT3 K429E or a known crenolanib-sensitive variant, such as FLT3 D835Y, and test crenolanib sensitivity against both conditions (which is the new experiment we performed and included). To address questions about the controls, we have now repeated the experiment and included additional controls, using FLT3 WT and empty vector transduced cells, which are not sensitive to crenolanib. As expected, crenolanib is not cytotoxic to FLT3 WT or empty vector transduced cells even at high concentrations (**Revision3 figure 1e**). In addition, we have included another crenolanib-sensitive condition, FLT3-ITD-transduced bone marrow cells. As expected, FLT3-ITD- and FLT3-D835-transduced bone marrow cells are sensitive to crenolanib (**Revision3 figure 1e**). FLT3-K429E-transduced bone marrow cells are significantly less sensitive to crenolanib. Therefore, we have shown that FLT3 K429E transduced bone marrow cells are less sensitive to FLT3 D835Y or FLT3-ITD transduced bone marrow cells. Please also find these new data in Supplementary figure 1a.*

*In all other instances, the question being asked is how **additional** mutant genes contribute to drug resistance in the context of mutant – not WT – FLT3. In this case, the appropriate controls are to compare the mutant versus the WT setting of these **additional** genes co-expressed with mutant FLT3. These comparisons are included throughout the manuscript, such as in Figures 3B, 3C, 4C, 4D, 5B, 5C, 5D, 5F, 5G.*

References

1. Smith, C. C. *et al.* Crenolanib is a selective type I pan-FLT3 inhibitor. *Proc. Natl. Acad. Sci.* **111**, 5319–5324 (2014).
2. Albers, C. *et al.* The secondary FLT3-ITD F691L mutation induces resistance to AC220 in FLT3-ITD + AML but retains in vitro sensitivity to PKC412 and Sunitinib. *Leukemia* **27**, 1416–1418 (2013).
3. Daver, N. *et al.* Secondary mutations as mediators of resistance to targeted therapy in

leukemia. *Blood* **125**, 3236–3245 (2015).

REVIEWERS' COMMENTS:

Reviewer #1 (Remarks to the Author):

The authors studied the role of Flt3 mutation in AML. However, experimental issues raised previously have not been address; compromising the interpretation of results and the importance of the findings.

- Use of Ba/F3 cell line. This cell line is a murine proB cell line. It is not human and not even myeloid. Even though it is a nice system to study the implication of flt3, results have to be validated in a human myeloid system. Moreover, comparing BA/F3 murine B cells with the human myeloid FLT3-ITD positive Molm-14 cell line is of limited interest due to important different between systems. Fig 1B-E.

- Lack of transfection controls. Figure 3B. There is no Wt control transfected (or empty vector control). Again, the muring B cell line Ba/F3.

- Use of murine BM cells. Which type of cells have been analysed? Bulk bone marrow cells (myeloid + lymphoid + erythroid + mega + primitive cells)? Which type of CFUs? Blasts? Myeloid Again, an extraordinary heterogenous murine blood population. Taking into account the localization of the laboratory and the research programs conducted there, access to human blood cells should be feasible. Indeed, it is not difficult to obtain healthy blood samples. Also, it is not difficult to isolate more homogenous populations. I agree that the question is the sensitivity of a mutation to the treatment with crenolanib but the environment of the experiments is crucial for the question to be answered.

Reviewer #1 (Remarks to the Author):

The authors studied the role of Flt3 mutation in AML. However, experimental issues raised previously have not been addressed; compromising the interpretation of results and the importance of the findings.

- Use of Ba/F3 cell line. This cell line is a murine proB cell line. It is not human and not even myeloid. Even though it is a nice system to study the implication of flt3, results have to be validated in a human myeloid system. Moreover, comparing BA/F3 murine B cells with the human myeloid FLT3-ITD positive Molm-14 cell line is of limited interest due to important differences between systems. Fig 1B-E.

Answer: we have expressed FLT3 WT and different mutants in Ba/F3 cells, murine lineage negative stem cells, and the human leukemia cell line Molm14 cells and have shown consistently that FLT3 K429E is less sensitive to crenolanib comparing to FLT3-ITD or FLT3 D835Y.

- Lack of transfection controls. Figure 3B. There is no Wt control transfected (or empty vector control). Again, the murine B cell line Ba/F3.

Answer: the question asked here is whether a co-occurred PTPN11 mutation confers crenolanib resistance to FLT3 D835Y mutation. Notably, crenolanib was not aimed to treat FLT3 WT leukemia patients. In addition, FLT3 WT, PTPN11 WT or mutant, or FLT3 WT/PTPN11 WT or mutant cells do not transform Ba/F3. Therefore, we believe the appropriately controlled experiment is the one that was performed – evaluating drug sensitivity of FLT3 D835Y alone or in combination with PTPN11 WT or mutant.

- Use of murine BM cells. Which type of cells have been analysed? Bulk bone marrow cells (myeloid + lymphoid + erythroid + mega + primitive cells)? Which type of CFUs? Blasts? Myeloid Again, an extraordinary heterogeneous murine blood population. Taking into account the localization of the laboratory and the research programs conducted there, access to human blood cells should be feasible. Indeed, it is not difficult to obtain healthy blood samples. Also, it is not difficult to isolate more homogeneous populations. I agree that the question is the sensitivity of a mutation to the treatment with crenolanib but the environment of the experiments is crucial for the question to be answered.

Answer: we have used murine BM lineage negative cells which represent a relatively homogeneous population of BM stem cells. The CFU assay is a myeloid differentiation cytokine (GF3534) cocktail, which is commonly used to evaluate myeloid leukemic potential.

The only suitable human healthy blood samples are CD34+ stem cells, which are normally only seen in the BM with less than 1% frequency. Therefore, it's very difficult to get access. The murine lineage negative stem cells we have used in this experiment are equivalent to human CD34+ stem cells.

Again, we have used three different models – Ba/F3 cells, murine lineage negative stem cells, and the human leukemia cell line Molm14 cells with consistent results. We believe these data are sufficient to show that FLT3 K429E is less sensitive to crenolanib compared with FLT3-ITD or FLT3 D835Y.